# Decomposing the Enigma: Subgoal-based Demonstration Learning for Formal Theorem Proving

## Abstract

Large language models (LLMs) present an intriguing avenue of exploration in the domain of formal theorem proving. Nonetheless, the full utilization of these models, particularly in terms of demonstration formatting and organization, remains an underexplored area. In an endeavor to enhance the efficacy of LLMs, we introduce a subgoal-based demonstration learning framework, consisting of two primary elements: Firstly, drawing upon the insights of subgoal learning from the domains of reinforcement learning and robotics, we propose the construction of distinct subgoals for each demonstration example and refine these subgoals in accordance with the pertinent theories of subgoal learning. Secondly, we build upon recent advances in diffusion models to predict the optimal organization, simultaneously addressing two intricate issues that persist within the domain of demonstration organization: subset selection and order determination. Through the integration of subgoal-based learning methodologies, we have successfully increased the prevailing proof accuracy from 38.9% to 44.3% on the miniF2F benchmark. Furthermore, the adoption of diffusion models for demonstration organization can lead to an additional enhancement in accuracy to 45.5%, or a $5\times$ improvement in sampling efficiency compared with the long-standing state-of-the-art method.[1]

## 1 Introduction

Mathematical theorem proving constitutes a significant milestone in the pursuit of artificial intelligence. Recently, machine learning methodologies have spurred advancements in both formal and informal theorem proving domains (Polu & Sutskever, 2020; Lewkowycz et al., 2022). Our study falls into the former category. In contrast to informal theorem proving, formal methods have the advantage of leveraging interactive proof assistants (Paulson, 2000) to automatically validate proofs generated by models, delegating the verification task to computational systems rather than human intervention. This significantly reduces the costs associated with proof checking, and has been applied in software verification (Klein et al., 2009) and research-level mathematics (Castelvecchi et al., 2021).

Recently, advances in large language models (LLMs) shed new light on the domain of formal theorem proving. The complexity of automated theorem proving comes from the necessity of searching through a vast space of possible logical statements and proof methods, in order to determine the truth-value of a given theorem. LLMs reduce the difficulty of the searching problem by factorizing the formal proof automation task into two in-context learning (§5.2) problems (Wu et al., 2022a; Jiang et al., 2022a; First et al., 2023). Given a mathematical *statement*, an LLM first generates its *informal proof* as a draft. It then generates a *formal sketch* based on this draft, which is ready for an off-the-shelf prover to verify its correctness automatically.[2] In both of these steps, the quality of the demonstrative in-examples either written by humans or generated by machines is the key to the performance of the system.

---

[1]All data and code will be open-sourced upon publication.

[2]In practice, the *informal proof* often serves as inline comments in the *formal sketch* to better guide the generation procedure.

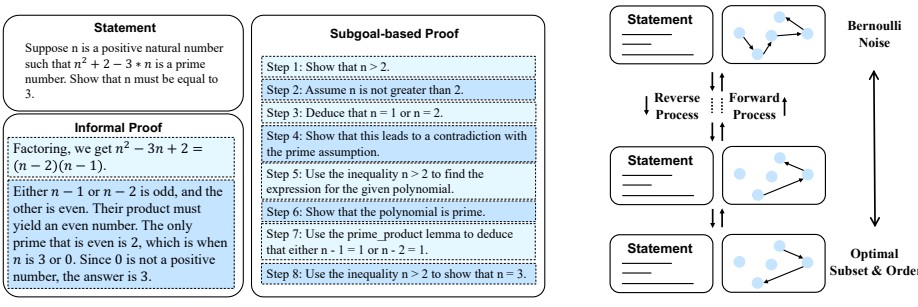

(a) Subgoal-based Proof        (b) Demonstration Reorganization

Figure 1: **Left**: An instance of informal proof and subgoal-based proof. **Right**: Employing diffusion models to identify a more effective subset of demonstration examples, as well as the optimal order for these examples.

In this paper, we seek to improve the efficacy of LLMs in formal theorem proving by delving deeper into the format and the organization of these demonstrative in-context examples. We present a subgoal-based demonstration learning framework, comprising two main components. First, we restructure an *informal proof* into a *subgoal-based proof* (Figure 1(a)), drawing upon the insights of subgoal learning from reinforcement learning and robotics, where studies show that breaking down complex tasks into smaller yet more uniformed subgoals enhances the learning efficiency of the agents (Eysenbach et al., 2019; Zhang et al., 2021). To construct subgoal-based proofs that can be easily processed and handled by LLMs, we start with human-written informal proofs and then iteratively refine them through interaction with ChatGPT (Team, 2022), guided by the subgoal learning theory (§2.1). Second, a recent study (Wu et al., 2022b) points out that the selection and the ordering of the in-context examples have a significant impact on performance. The lengthy formal sketches in automatic theorem proving intensify this impact, as we can only present very few cases of demonstrations. In response to that, we train a diffusion model to organize the demonstrative in-examples for the translation process from *subgoal-based proof* to its corresponding *formal sketch* of each instance (§2.2). This approach identifies a more effective subset of demonstration examples as well as the most beneficial order of these examples (Figure 1(b)).

The proposed method significantly outperforms competing approaches in formal theorem proving tasks, achieving a pass rate of $45.5\%$ on miniF2F dataset (Zheng et al., 2021), a $6.6\%$ absolute and $17.0\%$ relative improvement over the previous state-of-the-art system (Jiang et al., 2022a). Furthermore, the adoption of diffusion models for demonstration selection and ordering can lead to a significant improvement in sampling efficiency, reaching previous state-of-the-art ($38.5\%$) on miniF2F with only $20$ (compared to $100$) calls to the LLM.

## 2   SUBGOAL-BASED DEMONSTRATION LEARNING

Given a theorem statement $x$, the goal of proof synthesis is to generate a formal sketch $y$ which can be verified by an off-the-shelf automated theorem prover (e.g., Sledgehammer) (Jiang et al., 2022a). In this section, we elaborate the subgoal-based demonstration learning framework that consists of two key components, subgoal-based proof (§2.1) and demonstration reorganization (§2.2). The *subgoal-based proof* replaces the *informal proof*, breaking down a complex problem into smaller subgoals that offer more fine-grained and uniform guidance to the LLMs. The *demonstration reorganization* takes place in the stage of generating the *formal sketch* based on the *subgoal-based proof*. This procedure is non-trivial. Given the limited context length of the LLMs, selecting relevant yet diverse demonstration examples has a significant impact on the final pass rate of these formal sketches. We denote the set of all $N$ demonstration examples by $E = \{E_1, E_2, \cdots, E_N\}$. Each of them contains a mathematical *statement*, an *informal proof* (or a *subgoal-based proof*), and a *formal sketch*.[3] In the remainder of this section, we first describe the iterative refinement process that produces the subgoal-based proofs given the informal proof, guided by the principles in the subgoal learning theory (Zhang et al., 2021). We then explain our solution to the demonstration reorganization. Starting from collecting

---

[3] A more comprehensive illustration of these terms is provided in Figure 3.

arrangements that have yielded successful proofs, we use these as training data for a diffusion model, which progressively determines the most favorable reorganization during inference.

## 2.1 SUBGOAL-BASED PROOF

The significance of LLMs to formal theorem proving is that they grant us the ability to leverage informal proofs to guide formal theorem proving, which otherwise has to be based on expensive heuristics-based brute-force search. Despite considerable progress (Lewkowycz et al., 2022; OpenAI, 2023), this approach suffers from the flawed informal proofs generated by the LLMs (Jiang et al., 2022a). We propose to use subgoal-based proofs to replace the informal proofs, where the subgoals are strictly aligned with the states in the automatic provers. Following Zhang et al. (2021), we seek to obtain a *valid* sequence of subgoals which satisfies the condition that each subgoal in this sequence should be reachable from the initial state (i.e., the statement) and attain the final state (i.e., the passing state of the proof). These valid sequences integrate the guidance from the LLMs better with the search space of the automatic theorem provers, thereby leveraging the ability of the LLMs to the maximum extent. However, it is non-trivial to get these valid subgoal proofs as human-written subgoals often fall short of the above constraints. To address this problem, we iteratively refine the subgoal proof, in the spirit of self-play in reinforcement learning (Silver et al., 2016), making calls to both the LLM and the off-the-shelf automated theorem prover.

**Subgoal Refinement.** We start with manually written subgoal-based proofs, and denote these as the initial seed set $\{E_i^{(0)}\}_{i=1}^N$. This set contains subgoal-based proofs formed on the informal proofs and the statement, yet not guaranteed to be a valid sequence. We denote the sequence of subgoals in an instance as $(s_0, s_1, s_2, \cdots, s_\Delta, s_{\Delta+1})$, where $\Delta$ is the total number of subgoals and $s_0$ and $s_{\Delta+1}$ are two special subgoals that align with the initial and final states of the automatic prover. During the $k$-th iteration, we randomly select a subset of instances from the previous iteration $\{E_i^{(k-1)}\}_{i=1}^N$ as the in-context demonstration for the LLM to generate subgoals for a given instance. According to the definition, $s_i$ is considered to be a valid subgoal if and only if it can be reached from $s_0$ and can reach $s_{\Delta+1}$. Therefore, for each of the subgoal, we recursively call the proof assistant to verify the validness of the most recently developed subgoal and only after $\Delta$ recursions we can obtain the new valid sequence of subgoals and adds that into the next iteration as $E_i^{(k)}$. This process improves the consistency of the derived subgoals in style, thus making it easier for the LLM to learn from in the inference stage. We provide a detailed description of the algorithm in the Appendix.

## 2.2 DEMONSTRATION REORGANIZATION

The demonstration examples can be lengthy in formal theorem proving. If we assume a maximum context length of 3072 tokens, only $4.79$ examples on average can be included. Our experiments echo the findings by Wu et al. (2022b). These instance-based demonstration examples have a significant impact on performance. Only certain orders of carefully chosen demonstration examples lead to successful theorem proving. Consequently, identifying the optimal subset from the pool and ordering them into meaningful in-context demonstration examples is of great significance, which unfortunately is an NP-complete problem. We formulate the demonstration reorganization problem as finding the (Sub)hamiltonian graph where the nodes represent demonstration examples, and a traverse following the path corresponds to the selection and ordering of them. Building upon the recent success of applying diffusion models in addressing NP-complete problems (Graikos et al., 2022; Sun & Yang, 2023), we further treat this problem as a diffusion process on the graph. This solution has two main advantages. First, it addresses the example selection and ordering problem simultaneously. Second, the inference can be performed in parallel, which greatly reduces the time of discovering the optimal arrangement given the demonstration examples. We start from collecting successful pairs of in-context demonstration example organization and the corresponding statement $x$ as the training data for the diffusion model. We randomly organize (select and order) the demonstration examples and query the LLM to see if it can generate the proof successfully. The passing cases will be used as the starting configuration $\psi_0$ in the diffusion process given the statement $x$.

**Training.** The aim of employing diffusion models is to predict the optimal organization, denoted as $\psi_0$,[4] conditioning on the theorem statement $x$. From the standpoint of variance inference, diffusion

---

[4] $\psi_0$ represents the sequence of demonstrations that, when fed into the prompt, results in a successful formal sketch.

models adopt the following formulations to model $p_{\boldsymbol{\theta}}(\boldsymbol{\psi}_0|x)$,

$$p_{\boldsymbol{\theta}}(\boldsymbol{\psi}_0|x) \coloneqq \int p_{\boldsymbol{\theta}}(\boldsymbol{\psi}_{0:T}|x)\mathrm{d}\boldsymbol{\psi}_{1:T}, \tag{1}$$

where $\boldsymbol{\psi}_1, \cdots, \boldsymbol{\psi}_T$ serve as latent variables with the same dimensionality as $\boldsymbol{\psi}_0$. The learned reverse process progressively denoises these latent variables in order to reconstruct $\boldsymbol{\psi}_0$. This procedure can be formalized as follows,

$$p_{\boldsymbol{\theta}}(\boldsymbol{\psi}_{0:T}|x) = p(\boldsymbol{\psi}_T)\prod_{t=1}^{T} p_{\boldsymbol{\theta}}(\boldsymbol{\psi}_{t-1}|\boldsymbol{\psi}_t, x). \tag{2}$$

The forward process gradually corrupts $\boldsymbol{\psi}_0$ to generate noised latent variables,

$$q(\boldsymbol{\psi}_{1:T}|\boldsymbol{\psi}_0) = \prod_{t=1}^{T} q(\boldsymbol{\psi}_t|\boldsymbol{\psi}_{t-1}). \tag{3}$$

The goal of the training process is to maximize the evidence lower bound (ELBO),

$$
\begin{aligned}
\mathbb{E}\left[\log p_{\boldsymbol{\theta}}(\boldsymbol{\psi}_0|x)\right] &\geq \mathbb{E}_q\left[\log \frac{p_{\boldsymbol{\theta}}(\boldsymbol{\psi}_{0:T}|x)}{q_{\boldsymbol{\theta}}(\boldsymbol{\psi}_{1:T}|\boldsymbol{\psi}_0, x)}\right] \\
&= \mathbb{E}_q\left[\log p_{\boldsymbol{\theta}}(\boldsymbol{\psi}_0|\boldsymbol{\psi}_1, x) - \sum_{t>1} D_{\mathrm{KL}}[q(\boldsymbol{\psi}_{t-1}|\boldsymbol{\psi}_t, \boldsymbol{\psi}_0)\|p_{\boldsymbol{\theta}}(\boldsymbol{\psi}_{t-1}|\boldsymbol{\psi}_t, x)]\right].
\end{aligned} \tag{4}
$$

We employ a Graph Neural Network (GNN) for the encoding and denoising process of the graph. Following Austin et al. (2021), we adopt discrete diffusion models to model binary random variables.

**Inference.** During the inference stage, we obtain samples $\boldsymbol{\psi} \sim p_{\boldsymbol{\theta}}(\boldsymbol{\psi}_0|x)$ and subsequently reconstruct the order of demonstration examples from $\boldsymbol{\psi}$. We then incorporate examples sequentially into the LLM context, and define the output of the demonstration organization module as the sequence of examples upon reaching the LLM length constraint. More details of the implementation of the diffusion model, the implementation of GNN, and techniques used in the sampling process of $\boldsymbol{\psi}$ can be found in the Appendix.

## 3 EXPERIMENTS

### 3.1 FORMAL ENVIRONMENT

**Interactive Theorem Provers.** Interactive Theorem Provers (ITPs), such as Isabelle (Paulson, 1994), constitute the backbone of contemporary mathematical verification systems. They facilitate the integration of mathematical definitions and theorems into a consistent logical framework, such as Higher-Order Logic or Dependent Type Theory, which is operationalized by their kernels. The kernel plays a pivotal role in the verification process, meticulously examining each theorem to ascertain its recognition by the ITP and thereby ensuring the integrity of the system. The theorem proving process within an ITP is characterized by the articulation of the theorem in the ITP's programming language, followed by an iterative simplification into more manageable objectives or subgoals. The theorem is deemed proven once it can be distilled down to pre-established facts. The selection of Isabelle for our paper is motivated by its intuitive interface, its compatibility with a range of logical frameworks, and its comprehensive library of formalized mathematics.

**Sledgehammer.** Sledgehammer (Paulsson & Blanchette, 2012) serves as a powerful tool for automating reasoning within the interactive theorem prover Isabelle. It functions by transmuting the goals encapsulated in Isabelle/HOL's higher-order logic into alternative logics, such as first-order logic. These transmuted goals are then passed to off-the-shelf automated theorem provers, including E, CVC4, Z3, Vampire, and SPASS. In the event that any of these automated theorem provers successfully derives the proof in their respective formats, Sledgehammer undertakes the task of reconstructing the proof within the Isabelle/HOL framework using certified provers, namely metis, meson, and smt. This reconstructed proof, being more interpretable to humans, significantly enhances the system's usability, thereby contributing to the efficiency and effectiveness of (interactive) theorem proving.

### 3.2 DATASET AND EVALUATION

**Dataset.** We evaluate our approach using the miniF2F dataset (Zheng et al., 2021), which comprises 488 formal mathematical problems derived from high-school competitions, expressed in three formal languages: Lean, HOL-Light, and Isabelle. The dataset is divided into a validation and a test set, each including 244 problems. The problems within the dataset are sourced from three distinct categories: 260 problems are extracted from the MATH dataset (Hendrycks et al., 2021), 160 problems are extracted from actual high-school mathematical competitions (AMC, AIME, and IMO), and 68 problems are crafted to mirror the difficulty level of the aforementioned competitions.

**Evaluation.** The task at hand entails the generation of formal sketches for problems in the miniF2F dataset. The validity of a formal sketch depends on two criteria: first, the absence of "cheating" keywords such as "sorry" and "oops" that prematurely terminate a proof prior to its completion; second, the capacity of the interactive theorem prover Isabelle to authenticate the corresponding formal statement with the proof. To make working with Isabelle easier, we use the Portal-to-Isabelle API, as introduced by Jiang et al. (2022a). Given the absence of a training split in the miniF2F dataset, we leverage optimal organizations that yield successful proofs from the miniF2F-valid set to train the diffusion model. As proposed by Lample et al. (2022), we employ the cumulative pass rate as a measure for the results obtained from performing inference using diffusion models on the miniF2F-valid set. This involves integrating the pass rates from both the data collection stage for training and the inference stage. When it comes to other scenarios, namely conducting inference on the miniF2F-test or cases where the diffusion model is not employed, we simply provide the pass rate.

### 3.3 BASELINES

We leverage the following baselines to substantiate the effectiveness of our proposed methodology:

**Symbolic Automated Provers.** We first employ Sledgehammer, a proof automation tool that is extensively utilized within the Isabelle environment. We adhere to the default configuration of Sledgehammer as provided in Isabelle2021, which encompasses a 120-second timeout and a suite of five automated theorem provers (Z3, CVC4, SPASS, Vampire, E). In alignment with Jiang et al. (2022a), we employ Sledgehammer supplemented with heuristics, integrating 11 prevalent tactics (i.e., auto, simp, blast, fastforce, force, eval, presburger, sos, arith, linarith, auto simp: field simps) with Sledgehammer. If all the tactics fail or take longer than 10 seconds, the system reverts to the base Sledgehammer.

**Search-based Methods.** In addition to the above, we incorporate baselines that utilize Monte-Carlo tree search (Silver et al., 2016) to discover the proof. This includes Thor (Jiang et al., 2022b) and another version of Thor that employs an expert iteration on autoformalized data (i.e., Thor+expert iteration (Wu et al., 2022a)). Thor combines language models with automatic theorem provers to overcome the challenge of selecting beneficial premises from a vast library. Thor+expert iteration enhances a neural theorem prover by training it on theorems that have been automatically formalized.

**LLM-based Method.** Lastly, we incorporate an LLM-based baseline, Draft, Sketch and Prove (DSP) (Jiang et al., 2022a). DSP turns informal proofs into formal sketches and leverages these formal sketches to steer an automated prover. Notably, we employ the variant of DSP that is implemented with the $540B$ Minerva model (Lewkowycz et al., 2022), as this particular implementation demonstrated superior performance in their paper. Similar to our method, this DSP variant also uses the Sledgehammer tool, maintaining consistency in tool usage across both approaches.

We exclude representative methods such as HyperTree Proof Search (HTPS) (Lample et al., 2022) and GPT-f with expert iteration (Polu et al., 2022), which are implemented using Lean (de Moura et al., 2015), a different interactive theorem prover. The disparity in tactics and automation between Lean and Isabelle renders them not directly comparable to our method.

### 3.4 IMPLEMENTATION DETAILS

Throughout our work, we employ ChatGPT (i.e., the *gpt-3.5-turbo-0301* version) as the LLM. For the creation of the formal sketch, the temperature and max_tokens parameters of *gpt-3.5-turbo-0301*

Table 1: Pass rates on the miniF2F dataset with Isabelle. Numbers in bold denote the best performance. Numbers with a ⋆ correspond to the cumulative pass rate (Lample et al., 2022) since the evaluated statements are part of the training for diffusion models. See §3.2 for more details about cumulative pass rate.

|  | valid | test |
|---|---|---|
| Sledgehammer | 9.9% | 10.4% |
| Sledgehammer+heuristic | 18.0% | 20.9% |
| Thor | 28.3% | 29.9% |
| Thor + expert iteration | 37.3% | 35.2% |
| DSP (540B Minerva) | 42.6% | 38.9% |
| Ours | **48.0%**⋆ | **45.5%** |

Table 2: Ablation results on the miniF2F dataset with Isabelle. Numbers with a ⋆ correspond to the cumulative pass rate.

|  | valid | test |
|---|---|---|
| Ours | 48.0%⋆($\pm$0.4) | 45.5%($\pm$0.6) |
| - subgoal & diffusion | 41.4%($\pm$0.9) | 38.7%($\pm$1.2) |
| - subgoal | 44.3%⋆($\pm$0.7) | 40.6%($\pm$0.6) |
| - diffusion | 46.9%($\pm$1.3) | 44.1%($\pm$0.9) |

are set to $0$ and $1024$, respectively. For each subgoal proof, we make one formal sketch attempt, as suggested by previous literature (Jiang et al., 2022a). In terms of the establishment of the subgoal-based proof, we set the number of refinement iterations to be $15$, with the number of demonstration examples, denoted as $N$, being set to $61$. For demonstration organization, we employ a randomized demonstration organization approach to generate proofs for $116$ distinct statements on miniF2F-valid, which yield $137$ successful proofs. We then partition the corresponding demonstration contexts into a training set and a validation set, comprising $81$ and $56$ instances respectively. The training of our diffusion models is conducted with a learning rate of $5e - 4$, a batch size of $16$, and over a span of $50$ epochs. We set the number of diffusion steps, represented as $T$, to $80$. We employ an early stopping strategy on the validation set and report the performance averaged in three repetitive experiments.

## 3.5 MAIN RESULTS

The experiment results, as shown in Table 1, yield several key observations: (1) Our proposed method achieves a pass rate of 48.0% on miniF2F-valid and 45.5% on miniF2F-test, surpassing all competing methods. This superior performance is attributable to our novel application of diffusion models for demonstration reorganization coupled with subgoal-based proof; (2) The methods Thor and Thor + expert iteration struggle due to the enormously large action space. This space significantly overshadows that of games, thereby posing challenges to the comprehensive utilization of the Monte Carlo tree search. Consequently, these methods underperform when compared to LLM-based methods; and (3) DSP has pioneered the introduction of the informal proof, a critical step in the LLM-based formal theorem proving task. However, human-written informal proofs do not offer optimal compatibility with large language models. Our method, grounded in the subgoal-learning theory, is capable of inferring subgoal-based proofs that are more amenable to large language models.

## 4 ANALYSIS

### 4.1 ABLATION STUDY

In our ablation study, we examine four variations of our model on the miniF2F dataset, as detailed in Table 2. The models include our full method (Ours), and three variants: one excluding the subgoal-based proof, denoted as "-subgoal", which utilizes informal proofs instead; one without the diffusion method, referred to as "-diffusion", employing a randomized selection for demonstrations; and one, denoted as "-subgoal & diffusion", representing the basic form of *gpt-3.5-turbo*, where both the subgoal-based proof and diffusion method are omitted, functioning without subgoal-based proofs or diffusion models. To quantify the uncertainty associated with our method, we have supplemented

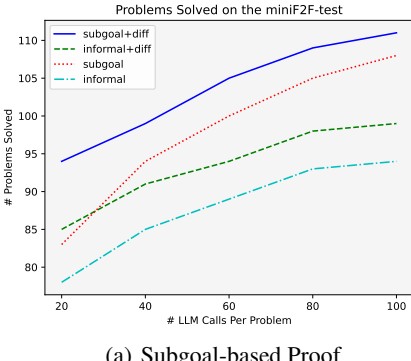
(a) Subgoal-based Proof

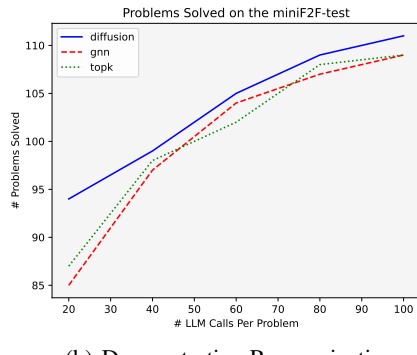
(b) Demonstration Reorganization

Figure 2: Number of problems solved on miniF2F-test against the number of LLM calls per problem, where the x-axis represents the number of independent problem-solving attempts. **Left**: a comparative assessment between the informal proof and subgoal-based proof under two distinct conditions: presence and absence of the diffusion model. **Right**: a comparative exploration of different in-context learning methods.

our results with standard deviation values. For the variants employing diffusion models ("Ours" and "-subgoal"), we trained three separate diffusion models as detailed in §3.4. For the variants without diffusion models ("'-subgoal & diffusion" and "-diffusion"), the experiment was repeated three times.

Our full model achieves the highest performance on both validation and test sets. This underscores the importance of both subgoal-based proof and demonstration reorganization. The model without both components showed the lowest performance, further emphasizing the significance of these components. The models missing either the subgoal-based proof or reorganization components also show decreased performance, indicating the substantial role of each component. Additionally, it is noteworthy that the methods incorporating diffusion models exhibit relatively more stable performance, affirming the stability benefits of these models in our study.

### 4.2 ON THE EFFECT OF SUBGOAL-BASED PROOF

We further use four different variants to explore the impact of subgoal-based proof. Figure 2(a) displays the results of this experiment, where "informal" denotes the utilization of informal proofs instead of subgoal-based proof, and "diff" indicates the integration of demonstration reorganization. The results indicate a significant difference between the approaches that incorporate subgoal-based proof ("subgoal" and "subgoal+diff") and those that do not ("informal" and "informal+diff"). This trend remains consistent across all LLM call numbers, suggesting a noteworthy positive effect of subgoal-based proof on the overall performance of our method.

### 4.3 ON THE EFFECT OF DEMONSTRATION REORGANIZATION

To further investigate the effect of a diffusion model for demonstration reorganization, we draw a comparative analysis between its performance and two alternative in-context learning methodologies: the Graph Neural Network (GNN) and the Top-K. The Top-K method calculates Ada embeddings for test problem statements and in-context examples, subsequently selecting the top-k most similar examples, as measured by dot product, ensuring those with higher similarity are positioned nearer to the test problem in the prompt. The GNN is congruent with a modified version of our proposed model when the inference diffusion step is set to 1, while the efficacy of the Top-K methodology has been extensively substantiated in the literature (Liu et al., 2021). Figure 2(b) presents the empirical results, manifesting that the diffusion model's performance increment diminishes as the number of LLM calls escalates to 100. This phenomenon stems from the fact that the module is trained on data collated from successful proofs via randomized organization sampling. Consequently, it may encounter difficulties in discerning the optimal organization for data that deviates significantly from its training dataset. Nevertheless, this limitation does not overshadow the potential of diffusion models to economize the number of LLM calls. Notably, with demonstration reorganization, our method exhibits an impressive capability of successfully deriving proofs for 94 problems (equivalently, a pass

---

**Statement:** Suppose n is a natural number such that n leaves a remainder of 3 when divided by 7. Show that (2n+1) leaves a remainder of 0 when divided by 7.

---

**Subgoal-based Proof:**
Step 1: Express n as $7k + 3$ for some k ∈ ℕ
Step 2: Substitute the expression for $n$ into the expression for $2n + 1$ and simplify.
Step 3: Show that the resulting expression is divisible by 7.

---

**Formal Sketch:**
```
theorem mathd_numbertheory_370:
  fixes n :: nat
  assumes h0 : "n mod 7 = (3::nat)"
  shows "(2*n+1) mod 7 = (0::nat)"
proof -
  (* Step 1: Express n as 7k + 3 for some k ∈ ℕ. *)
  have "n = 7 * (n div 7) + (n mod 7)"
    using div_mult_mod_eq[of "n" "7"] sledgehammer
  also have "... = 7 * (n div 7) + 3"
    using h0 sledgehammer
  finally have c0: "n = 7 * (n div 7) + 3".
  (* Step 2: Substitute the expression for n into the expression for 2n + 1 and simplify. *)
  have "2*n+1 = 2*(7*(n div 7) + 3) + 1"
    using c0 sledgehammer
  also have "... = 14*(n div 7) + 7"
    sledgehammer
  also have "... = 7*(2*(n div 7) + 1)"
    sledgehammer
  finally have c1: "2*n+1 = 7*(2*(n div 7) + 1)".
  (* Step 3: Show that the resulting expression is divisible by 7.*)
  show "(2*n+1) mod 7 = 0"
    using c1 sledgehammer
qed
```

Figure 3: A formal sketch generated by our proposed method.

rate of 38.5%), with a mere 20 LLM calls. Remarkably, this result is comparable with that of the DSP method, which necessitates 5× the number of LLM calls.

### 4.4 CASE STUDY

To better comprehend the efficacy of our proposed method, we present a formal sketch of a problem that remains unproven by earlier state-of-the-art methods. As demonstrated in Figure 3, it is apparent that our strategy successfully decomposes the complex objective into three manageable subgoals, each solvable by the LLM. We provide additional comprehensive examples in the Appendix.

## 5 RELATED WORK

### 5.1 MACHINE LEARNING FOR FORMAL THEOREM PROVING

Machine learning-based formal theorem proving systems primarily fall into two categories: those focusing on proof search strategies and premise selection, and those harnessing Large Language Models (LLMs) for autoformalization and proof generation. The first category, represented by works like Expert Iteration (Polu et al., 2022) and PACT (Han et al., 2021), devise novel learning strategies to enhance proof search, extracting self-supervised data from kernel-level proof terms. Systems such as HyperTree Proof Search (HTPS) (Lample et al., 2022) and Thor (Jiang et al., 2022b) integrate language models with automated theorem provers, while Magnushammer (Mikuła et al., 2023) presents a transformer-based approach for premise selection. While these techniques have proven effective, they struggle with increasing computational costs as theorems grow more complex. The second category exploits the potential of LLMs in the formalization of mathematical proofs. Both Wu et al. (2022a) and Jiang et al. (2022a) demonstrate that LLMs can convert mathematical problems into formal specifications, with the latter utilizing these translations to guide an automated prover. Baldur (First et al., 2023) extends this approach by generating entire proofs at once and introducing a proof repair model to enhance proving power. However, these approaches have yet to fully leverage the power of LLMs due to a lack of emphasis on the format and organization of demonstration examples. Our work aims to address this gap by introducing a subgoal-based demonstration learning framework that refines the use of LLMs in formal theorem proving.

## 5.2 IN-CONTEXT LEARNING

In the field of In-Context Learning (ICL), research primarily focuses on two main areas: (1) the selection of in-context examples, and (2) the arrangement of these examples in the learning context. With regard to the first area, Liu et al. (2021) suggest a retrieval-based prompt selection method, offering a thoughtful alternative to random example selection. This method aims to find examples that are semantically similar to a test sample to form related prompts. Building on this idea, Rubin et al. (2021) propose an effective retrieval process for prompts, using a pre-trained language model. Sorensen et al. (2022) further the exploration by introducing a new way to select prompt templates that don't need labeled examples or direct access to the model. Instead, they choose the template that maximizes the mutual information between the input and the corresponding model output. Su et al. (2022) present a two-step framework that is efficient in annotation. It first selects a set of examples from unlabeled data, and then retrieves task examples from the annotated set during testing. Lastly, Agrawal et al. (2022) focus on creating strategies specifically for machine translation tasks, emphasizing the importance of the quality and domain of in-context examples, and warning against the negative effects of unrelated noisy examples. Works in the second area examine the significance of the order in which prompts are presented. Zhao et al. (2021) point out the instability in few-shot learning caused by the order of training examples and suggest a calibration method to tackle this. Lu et al. (2021) delve deeper into this analysis, demonstrating the sensitivity of prompt order in few-shot learning situations. Even though previous efforts have made remarkable progress in either choosing or sequencing in-context examples, our research sets a new precedent by combining both elements. In this paper, we step out of these isolated areas of concentration, looking into an approach based on diffusion models that effectively tackles both the challenges of the selection and ordering at the same time.

## 5.3 SUBGOAL LEARNING

Subgoal learning is a pivotal concept in reinforcement learning. It can enable AI systems to solve complex, long-horizon tasks more effectively. Crucially, theoretical analyses have shed light on key concepts including the computational benefits of rewarding subgoals (Zhai et al., 2022), the structure of Markov decision processes beneficial for hierarchical reinforcement learning (Wen et al., 2020), the complexity of optimal option selection for planning (Jinnai et al., 2019a), and the integration of temporal abstraction into RL (Fruit et al., 2017). Empirical analyses in this field mainly focus on subgoal exploration, subgoal generation for planning, and curriculum learning for subgoals. Subgoal exploration aims to find the optimal or efficient exploration of subgoals, employing a variety of strategies. These include minimizing cover time (Jinnai et al., 2019b), learning dynamical distances (Hartikainen et al., 2019), maximizing entropy (Pitis et al., 2020), and utilizing asymmetric self-play (OpenAI et al., 2021). Subgoal planning research encompasses diverse algorithms for improved decision-making. For example, SoRB (Eysenbach et al., 2019) uses RL to build a graph for subgoal sequences, DC-MCTS (Parascandolo et al., 2020) applies learned sub-goal proposals to partition tasks, PAIR (Li et al., 2022) combines online RL and offline supervised learning, and Moro et al. (2022) extend MCTS with Hindsight Experience Replay for goal-oriented planning. The research centered on curriculum learning proposes various techniques to create a learning curriculum that gradually intensifies subgoal complexity, thereby optimizing learning efficiency and effectiveness (Zhang et al., 2020; 2021). While there have been preliminary efforts to apply similar principles in the construction of prompts for LLMs (Khot et al., 2022), the deployment of subgoal learning theories to manage intricate tasks, such as formal theorem proving, remains largely unexplored. Our work pioneers the use of subgoal learning in this domain, with a focus on format and organization.

## 6 CONCLUSION

In this paper, we have developed a subgoal-based demonstration learning framework that significantly enhances LLMs' efficacy in formal theorem proving. Our approach combines insights from subgoal learning and diffusion models, effectively addressing the challenges of demonstration formatting and organization. As a result, we achieve a 17.0% relative improvement in proof pass rate on the miniF2F benchmark and a $5\times$ improvement in sampling efficiency. Our work lays the foundation for future endeavors in leveraging AI for generating, validating, and contributing novel insights to automated theorem proving.

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

# A   MORE DETAILS ABOUT SUBOGAL-BASED PROOF

We provide a detailed description on the subgoal refinement method (see §2.1) through Algorithm 1. In the $k$-th iteration, we construct demonstration examples $\{E_i^{(k)}\}_{i=1}^N$ using improved subgoal-based proofs. To construct $E_i^{(k)}$, we first extract the statement and formal sketch from $E_i^{(k-1)}$, then use an LLM to generate subgoals. Afterward, a Refine module is called to confirm the validity of the created subgoals and adjust any subgoals identified as infeasible.

A subgoal $s_\Delta$ is deemed "reachable" from $s_1$ if, for any $s_i$ where $1 < i \leq \Delta$, the ATP solvers can bridge the gap between $s_{i-1}$ and $s_i$, under the assumption that all preceding subgoals $s_j$ (where $j < i$) are true. This concept is crucial in cases where the proof structure necessitates the independent verification of lemma a, b from $s_1$, followed by the substantiation of $s_2$ utilizing both lemma a and b. In such scenarios, as per our definition of reachability, the verification of $(s_{i-1}, s_i)$ inherently presumes the truth of all prior subgoals, denoted as all $s_j$ with $j < i$, which are encapsulated within the context $C$ in VERIF_AND_CORRECT. Specifically, the verification of $(s_a, s_b)$ presupposes $s_1$ and $s_a$, and the verification of $(s_b, s_2)$ presupposes $s_1$, $s_a$, and $s_b$. If the ITP valid the steps, the subgoal is deemed reachable; otherwise, it is considered unattainable.

In practice, the reachability of subgoals is tested using the VERIFY function in Algorithm 4, which integrates both an Interactive Theorem Prover (ITP) and a Large Language Model (LLM). It amends the formal sketch for a subgoal, assigns the LLM to complete the missing segments, and then validates these with the ITP. If the ITP affirms the steps, the subgoal is considered reachable; otherwise, it is deemed unattainable. In instances where the LLM consistently fails to generate provable subgoals for the theorem prover, the VERIFY_AND_CORRECT function will revert to returning the original subgoal pair $(s_i, s_{i+1})$.

We present an example to elucidate this process further (see Figures 4 to 12).[5] As shown in Figure 4, the LLM creates two subgoals for the theorem *amc12a_2003_p4*, leading to $\{s_0, s_1, s_2, s_3\}$. Refining these subgoals involves calling verify_and_correct$(s_0, s_1)$ to improve the subgoal $s_1$. This is depicted in Figures 5 to 12. We first use the LLM to reconstruct the subgoal related to the first step, but this attempt fails (Figure 5). Then we break down the subgoal $s_1$ into three more detailed subgoals (Figure 6), each of which is then verified using the same LLM (Figures 7 to 9). Due to the unsuccessful reconstruction of the second subgoal (Figure 8), it is further broken down into two more specific subgoals (Figure 10). The last two subgoals pass the verification process successfully (Figures 11 and 12). Finally, the output of verify_and_correct$(s_0, s_1)$, namely $S^{0\to1}$, is defined as the set that includes the steps from 1 to 4 shown in Figure 12.

---

**Algorithm 1** Iterative Subgoal Refinement

| **Requires:** | EXTRACT | extraction of statement and formal sketch |
|---|---|---|
| | COMPOSE | composing of a statement, formal sketch |
| | | and subgoals to form a demonstration example |
| | INITIALIZE_SUBGOALS | generate subgoals with a LLM |

**function** ITERATIVE_REFINEMENT$(\{E_1^{(0)}, E_2^{(0)}, \cdots, E_N^{(0)}\})$
    **for** $k$ in $1, 2, \ldots, K$ **do**
        **for** $i$ in $1, 2, \ldots, N$ **do**
            $x, y \leftarrow$ EXTRACT$(E_i^{(k-1)})$
            $s_0, s_1, \cdots, s_\Delta, s_{\Delta+1} \leftarrow$ INITIALIZE_SUBGOALS$(x, y, E^{(k-1)})$
            $S^{0\to(\Delta+1)} \leftarrow$ REFINE$((s_0, s_{\Delta+1}, \{s_1, s_2, \cdots, s_\Delta\}))$
            $E_i^{(k)} \leftarrow$ COMPOSE$(x, y, S^{0\to(\Delta+1)})$
        **end for**
    **end for**
    **return** $\{E_1^{(K)}, E_2^{(K)}, \cdots, E_N^{(K)}\}$
**end function**

---

[5]To simplify the illustration, we leave out redundant demonstration examples.

---

**Algorithm 2** Refinement Algorithm

---

**Requires:**   VERIFY_AND_CORRECT     verify the validness of the subgoals and correct them
                                                   if necessary

  **function** REFINE($s_i$, $s_{j+1}$, $\{s_{i+1}, \cdots, s_j\}$)
      **if** i = j **then return** VERIFY_AND_CORRECT($s_i, s_{i+1}$)
      **end if**
      $S^{i \to i+1} \leftarrow$ REFINE$(s_i, s_{i+1}, \{\})$
      $S^{i+1 \to j+1} \leftarrow$ REFINE$(s_{i+1}, s_{j+1}, \{s_{i+2}, \cdots, s_j\})$
      **return** $S^{i \to i+1} \cup S^{i+1 \to j+1}$
  **end function**

---

---

**Algorithm 3** Verify and Correct

---

**Requires:**   VERIFY     verify if $s_{i+1}$ is reachable from $s_i$
             CORRECT     correct the pair of subgoals if necessary
             $M$         the maximum number of LLM calls, which is set to 10 in our experiments
             $C$         context including formal sketch and current subgoal-based proof

  **function** VERIFY_AND_CORRECT($s_i$, $s_{i+1}$, $C$)
      budget $\leftarrow M$
      $Q \leftarrow \varnothing$        ▷ A priority queue; lower pos values indicate proximity to the formal statement
      $Q$.PUSH$((-C.pos(s_{i+1}), (s_i, s_{i+1})))$  ▷ Push subgoals based on their distance to the formal statement
      valid_subgoals $\leftarrow []$
      **while** $Q \neq \varnothing$ and budget > 0 **do**
          _, $(s, s') \leftarrow Q$.POP()                    ▷ Retrieve subgoal closest to the formal statement
          budget $\leftarrow$ budget $- 1$                         ▷ LLM call consumed by Verify
          **if** VERIFY$(s, s', C)$ **then**
              valid_subgoals.APPEND$((s, s'))$
          **else**
              budget $\leftarrow$ budget $- 1$                         ▷ LLM call consumed by Correct
              subgoals $\leftarrow []$
              subgoals, $C \leftarrow$ CORRECT$(s, s', C)$     ▷ Fix subgoal $s'$ or generate granular subgoals, then update the context
              **for** $(s_j, s_{j+1})$ in subgoals **do**
                 $Q$.PUSH$((-C.pos(s_{j+1}), (s_j, s_{j+1})))$
              **end for**
          **end if**
      **end while**
      **if** valid_subgoals $\neq \varnothing$ **then**
          **return** valid_subgoals
      **else**
          **return** $[(s_i, s_{i+1})]$
      **end if**
  **end function**

---

---

**Algorithm 4** Verify

---

**Requires:**   ITP     Interactive Theorem Prover, i.e., Isabelle in our experiments
             LLM     Large Language Model, i.e., GPT-3.5-turbo in our experiments

  **function** VERIFY($s_i$, $s_{i+1}$, C)
      llm_output $\leftarrow$ LLM($C$.verify_prompt($s_i$, $s_{i+1}$, $C$))
      **if** ITP(llm_output) = pass **then**
          **return** True
      **else**
          **return** False
      **end if**
  **end function**

---

---

**Algorithm 5** Correct

---

**Requires:**      LLM      Large Language Model, i.e., GPT-3.5-turbo in our experiments
                      PARSE    Parse the output of LLM to extract subgoals
                                     which correspond to the same segment of the formal sketch as $s_{i+1}$
                   UPDATE    Replace $s_{i+1}$ in C with newly generated subgoals

  **function** CORRECT($s_i$, $s_{i+1}$, C)
      llm_output ← LLM($C$.correct_prompt($s_i$, $s_{i+1}$, $C$))
      subgoals ← PARSE(llm_output, $s_{i+1}$, $C$)
      C ← UPDATE(llm_output, $s_{i+1}$, $C$)
      **return** subgoals, $C$
  **end function**

---

Your task is to transform a formal proof, written in a style suitable for a theorem-proving system, into a more explanatory and narrative style, suitable for humans. The task involves adding explanatory comments that break down the formal proof into understandable steps, providing context and insight into the underlying logical structure and reasoning.

- - - - - - - - - - - - - - - - - - - - - - - - - - - - - - - - - - - - - - - - - - - - - - - - - - - - - - - - - - - - - - - - -

**Input:**
```
theorem amc12a_2003_p24:
    fixes a b::real
    assumes "b \<le>a" and "1<b"
    shows "ln (a/b) / ln a + ln (b/a) / ln b \<le>0" (is "?L \<le> _")
proof −
    define x y where "x=ln a" and "y=ln b"
    have "y>0"
        using \<open>b>1\<close> unfolding y_def using ln_gt_zero sledgehammer
    moreover have "x \<ge>y"
        using \<open>a\<ge>b\<close> unfolding x_def y_def using assms(2) sledgehammer
    ultimately have "x>0"
        sledgehammer
    have "?L = (x-y)/x + (y-x)/y"
        apply (subst (1 2) ln_div)
        using assms unfolding x_def y_def sledgehammer
    also have "... = 2 - (y/x + x/y)"
        using \<open>x>0\<close> \<open>y>0\<close> sledgehammer
    also have "... \<le> 0"
    proof −
        have "sqrt (x/y) * sqrt (y/x) = 1"
            sledgehammer
        have "0\<le> (sqrt (x/y) - sqrt (y/x))^2"
            sledgehammer
        also have "... = x / y - 2 + y / x"
            unfolding power2_eq_square using \<open>x>0\<close> \<open>y>0\<close> \<open>sqrt (x/y) * sqrt (y/x) = 1\<close>
            sledgehammer
        finally show ?thesis
            sledgehammer
    qed
    finally show ?thesis.
qed
```

- - - - - - - - - - - - - - - - - - - - - - - - - - - - - - - - - - - - - - - - - - - - - - - - - - - - - - - - - - - - - - - - -

**Output:**
```
theorem amc12a_2003_p24:
    fixes a b::real
    assumes "b \<le>a" and "1<b"
    shows "ln (a/b) / ln a + ln (b/a) / ln b \<le>0" (is "?L \<le> _")
proof −
    (* Step 1: With the definitions x = ln a, y = ln b, and the constraints y > 0 and x ≥ y, the equation simplifies into 2 − y/x − x/y. *)
    define x y where "x=ln a" and "y=ln b"
    have "y>0"
        using \<open>b>1\<close> unfolding y_def using ln_gt_zero sledgehammer
    moreover have "x \<ge>y"
        using \<open>a\<ge>b\<close> unfolding x_def y_def using assms(2) sledgehammer
    ultimately have "x>0"
        sledgehammer
    have "?L = (x-y)/x + (y-x)/y"
        apply (subst (1 2) ln_div)
        using assms unfolding x_def y_def sledgehammer
    also have "... = 2 - (y/x + x/y)"
        using \<open>x>0\<close> \<open>y>0\<close> sledgehammer
    (* Step 2: Using the inequality (√(x/y) − √(y/x))² ≥ 0, it is shown that the simplified equation 2 − y/x − x/y is less than or equal to 0. *)
    also have "... \<le> 0"
    proof −
        have "sqrt (x/y) * sqrt (y/x) = 1"
            sledgehammer
        have "0\<le> (sqrt (x/y) - sqrt (y/x))^2"
            sledgehammer
        also have "... = x / y - 2 + y / x"
            unfolding power2_eq_square using \<open>x>0\<close> \<open>y>0\<close> \<open>sqrt (x/y) * sqrt (y/x) = 1\<close>
            sledgehammer
        finally show ?thesis
            sledgehammer
    qed
    finally show ?thesis.
qed
```

Figure 4: Illustration of the $\mathrm{Initialize\_subgoals}$ function as denoted in Algorithm 1. ChatGPT is leveraged to generate the subgoal-based proof with respect to a formal sketch.

Your task is to complete the missing part of a Isabelle proof based on the corresponding description provided. The proof will consist of several steps, and you will need to use your knowledge of mathematical concepts and logical reasoning to fill in the gap.

**Input:**
```
theorem amc12a_2003_p24:
   fixes a b::real
   assumes "b \<le>a" and "1<b"
   shows "ln (a/b) / ln a + ln (b/a) / ln b \<le>0" (is "?L \<le> _")
proof -
   (* Step 1: With the definitions x = ln a, y = ln b, and the constraints y > 0 and x ≥ y, the equation simplifies into 2 − y/x − x/y. *)

   (* Step 2: Using the inequality (√(x/y) − √(y/x))² ≥ 0, it is shown that the simplified equation 2 − y/x − x/y is less than or equal to 0. *)
   also have "... \<le> 0"
   proof -
      have "sqrt (x/y) * sqrt (y/x) = 1"
         sledgehammer
      have "0\<le> (sqrt (x/y) - sqrt (y/x))^2"
         sledgehammer
      also have "... = x / y - 2 + y / x"
         unfolding power2_eq_square using \<open>x>0\<close> \<open>y>0\<close> \<open>sqrt (x/y) * sqrt (y/x) = 1\<close>
         sledgehammer
      finally show ?thesis
         sledgehammer
   qed
   finally show ?thesis.
qed
```

**Output:**
```
theorem amc12a_2003_p24:
   fixes a b::real
   assumes "b \<le>a" and "1<b"
   shows "ln (a/b) / ln a + ln (b/a) / ln b \<le>0" (is "?L \<le> _")
proof -
   (* Step 1: With the definitions x = ln a, y = ln b, and the constraints y > 0 and x ≥ y, the equation simplifies into 2 − y/x − x/y. *)
   define x y where "x=ln a" and "y=ln b"
   have "x > 0" and "y > 0"
      using assms by (simp_all add: ln_gt_zero)
   have "?L = 2 - y / x - x / y"
      using assms x_def y_def by (simp add: add_divide_distrib diff_divide_distrib ln_div)
   (* Step 2: Using the inequality (√(x/y) − √(y/x))² ≥ 0, it is shown that the simplified equation 2 − y/x − x/y is less than or equal to 0. *)
   also have "... \<le> 0"
   proof -
      have "sqrt (x/y) * sqrt (y/x) = 1"
         sledgehammer
      have "0\<le> (sqrt (x/y) - sqrt (y/x))^2"
         sledgehammer
      also have "... = x / y - 2 + y / x"
         unfolding power2_eq_square using \<open>x>0\<close> \<open>y>0\<close> \<open>sqrt (x/y) * sqrt (y/x) = 1\<close>
         sledgehammer
      finally show ?thesis
         sledgehammer
   qed
   finally show ?thesis.
qed
```

Figure 5: An instance of the "verify" component within the Verify_and_correct function in Algorithm 2. ChatGPT encounters a *failure* in reconstructing the proof associated with *step 1*, thereby deeming it an unsuitable subgoal.

The Isabelle Proof Assistant was unable to verify Step 1. To resolve this, consider providing more detailed information or clarity to enhance the explanation for this particular step.

--------------------------------------------------------------------------------

**Input:**
```
theorem amc12a_2003_p24:
   fixes a b::real
   assumes "b \<le>a" and "1<b"
   shows "ln (a/b) / ln a + ln (b/a) / ln b \<le>0" (is "?L \<le> _")
proof −
   (* Step 1: With the definitions x = ln a, y = ln b, and the constraints y > 0 and x ≥ y, the equation simplifies into 2 − y/x − x/y. *)
   define x y where "x=ln a" and "y=ln b"
   have "y>0"
      using \<open>b>1\<close> unfolding y_def using ln_gt_zero sledgehammer
   moreover have "x \<ge>y"
      using \<open>a\<ge>b\<close> unfolding x_def y_def using assms(2) sledgehammer
   ultimately have "x>0"
      sledgehammer
   have "?L = (x-y)/x + (y-x)/y"
      apply (subst (1 2) ln_div)
      using assms unfolding x_def y_def sledgehammer
   also have "... = 2 - (y/x + x/y)"
      using \<open>x>0\<close> \<open>y>0\<close> sledgehammer

   (* Step 2: Using the inequality (√(x/y) − √(y/x))^2 ≥ 0, it is shown that the simplified equation 2 − y/x − x/y is less than or equal to 0. *)
   also have "... \<le> 0"
   proof −
      have "sqrt (x/y) * sqrt (y/x) = 1"
         sledgehammer
      have "0\<le> (sqrt (x/y) - sqrt (y/x))^2"
         sledgehammer
      also have "... = x / y - 2 + y / x"
         unfolding power2_eq_square using \<open>x>0\<close> \<open>y>0\<close> \<open>sqrt (x/y) * sqrt (y/x) = 1\<close>
         sledgehammer
      finally show ?thesis
         sledgehammer
   qed
   finally show ?thesis.
qed
```

--------------------------------------------------------------------------------

**Output:**
```
theorem amc12a_2003_p24:
   fixes a b::real
   assumes "b \<le>a" and "1<b"
   shows "ln (a/b) / ln a + ln (b/a) / ln b \<le>0" (is "?L \<le> _")
proof −
   (* Step 1: Define x = ln a and y = ln b. *)
   define x y where "x=ln a" and "y=ln b"
   (* Step 2: With the conditions set as y > 0 and x ≥ y, we transition the equation to be framed in terms of x and y, thereby yielding (x−y)/x + (y−x)/y. *)
   have "y>0"
      using \<open>b>1\<close> unfolding y_def using ln_gt_zero sledgehammer
   moreover have "x \<ge>y"
      using \<open>a\<ge>b\<close> unfolding x_def y_def using assms(2) sledgehammer
   ultimately have "x>0"
      sledgehammer
   have "?L = (x-y)/x + (y-x)/y"
      apply (subst (1 2) ln_div)
      using assms unfolding x_def y_def sledgehammer
   (* Step 3: Simplify the expression to 2 − y/x − x/y. *)
   also have "... = 2 - (y/x + x/y)"
      using \<open>x>0\<close> \<open>y>0\<close> sledgehammer

   (* Step 4: Using the inequality (√(x/y) − √(y/x))^2 ≥ 0, it is shown that the simplified equation 2 − y/x − x/y is less than or equal to 0. *)
   also have "... \<le> 0"
   proof −
      have "sqrt (x/y) * sqrt (y/x) = 1"
         sledgehammer
      have "0\<le> (sqrt (x/y) - sqrt (y/x))^2"
         sledgehammer
      also have "... = x / y - 2 + y / x"
         unfolding power2_eq_square using \<open>x>0\<close> \<open>y>0\<close> \<open>sqrt (x/y) * sqrt (y/x) = 1\<close>
         sledgehammer
      finally show ?thesis
         sledgehammer
   qed
   finally show ?thesis.
qed
```

Figure 6: An instance of the "correct" component within the VERIFY_AND_CORRECT function in Algorithm 2. ChatGPT works on the decomposition of the original subgoal (i.e., step 1 in the input) into a series of more granular subgoals (i.e., step 1 - 3 in the output).

```
Your task is to complete the missing part of a Isabelle proof based on the corresponding description provided. The proof will consist of
several steps, and you will need to use your knowledge of mathematical concepts and logical reasoning to fill in the gap.
- - - - - - - - - - - - - - - - - - - - - - - - - - - - - - - - - - - - - - - - - - - - - - - - - - - - - - - - - - - - - - - - - - - - - - - - - - - - - - - - - - - - - -

Input:
theorem amc12a_2003_p24:
    fixes a b::real
    assumes "b \<le>a" and "1<b"
    shows "ln (a/b) / ln a + ln (b/a) / ln b \<le>0" (is "?L \<le> _")
proof –
    (* Step 1: Define x = ln a and y = ln b. *)

    (* Step 2: With the conditions set as y > 0 and x ≥ y, we transition the equation to be framed in terms of x and y, thereby yielding
    x−y / x + y−x / y. *)
    have "y>0"
        using \<open>b>1\<close> unfolding y_def using ln_gt_zero sledgehammer
    moreover have "x \<ge>y"
        using \<open>a\<ge>b\<close> unfolding x_def y_def using assms(2) sledgehammer
    ultimately have "x>0"
        sledgehammer
    have "?L = (x-y)/x + (y-x)/y"
        apply (subst (1 2) ln_div)
        using assms unfolding x_def y_def sledgehammer
    (* Step 3: Simplify the expression to 2 − y/x − x/y. *)
    also have "... = 2 - (y/x + x/y)"
        using \<open>x>0\<close> \<open>y>0\<close> sledgehammer

    (* Step 4: Using the inequality ( √(x/y) − √(y/x) )²≥ 0, it is shown that the simplified equation 2 − y/x − x/y is less than or equal to 0. *)

    also have "... \<le> 0"
    proof –
        have "sqrt (x/y) * sqrt (y/x) = 1"
            sledgehammer
        have "0\<le> (sqrt (x/y) - sqrt (y/x))^2"
            sledgehammer
        also have "... = x / y - 2 + y / x"
            unfolding power2_eq_square using \<open>x>0\<close> \<open>y>0\<close> \<open>sqrt (x/y) * sqrt (y/x) = 1\<close>
            sledgehammer
        finally show ?thesis
            sledgehammer
    qed
    finally show ?thesis.
qed
- - - - - - - - - - - - - - - - - - - - - - - - - - - - - - - - - - - - - - - - - - - - - - - - - - - - - - - - - - - - - - - - - - - - - - - - - - - - - - - - - - - - - -

Output:
theorem amc12a_2003_p24:
    fixes a b::real
    assumes "b \<le>a" and "1<b"
    shows "ln (a/b) / ln a + ln (b/a) / ln b \<le>0" (is "?L \<le> _")
proof –
    (* Step 1: Define x = ln a and y = ln b. *)
    define x y where "x=ln a" and "y=ln b"
    (* Step 2: With the conditions set as y > 0 and x ≥ y, we transition the equation to be framed in terms of x and y, thereby yielding
    x−y / x + y−x / y. *)
    have "y>0"
        using \<open>b>1\<close> unfolding y_def using ln_gt_zero sledgehammer
    moreover have "x \<ge>y"
        using \<open>a\<ge>b\<close> unfolding x_def y_def using assms(2) sledgehammer
    ultimately have "x>0"
        sledgehammer
    have "?L = (x-y)/x + (y-x)/y"
        apply (subst (1 2) ln_div)
        using assms unfolding x_def y_def sledgehammer
    (* Step 3: Simplify the expression to 2 − y/x − x/y. *)
    also have "... = 2 - (y/x + x/y)"
        using \<open>x>0\<close> \<open>y>0\<close> sledgehammer

    (* Step 4: Using the inequality ( √(x/y) − √(y/x) )²≥ 0, it is shown that the simplified equation 2 − y/x − x/y is less than or equal to 0. *)

    also have "... \<le> 0"
    proof –
        have "sqrt (x/y) * sqrt (y/x) = 1"
            sledgehammer
        have "0\<le> (sqrt (x/y) - sqrt (y/x))^2"
            sledgehammer
        also have "... = x / y - 2 + y / x"
            unfolding power2_eq_square using \<open>x>0\<close> \<open>y>0\<close> \<open>sqrt (x/y) * sqrt (y/x) = 1\<close>
            sledgehammer
        finally show ?thesis
            sledgehammer
    qed
    finally show ?thesis.
qed
```

Figure 7: An instance of the "verify" component within the Verify_and_correct function in Algorithm 2. ChatGPT *successfully* reconstructs the proof associated with *step 1*, thus validating it as a viable subgoal.

Your task is to complete the missing part of a Isabelle proof based on the corresponding description provided. The proof will consist of several steps, and you will need to use your knowledge of mathematical concepts and logical reasoning to fill in the gap.

- - - - - - - - - - - - - - - - - - - - - - - - - - - - - - - - - - - - - - - - - - - - - - - - - - - - - - - - - - - - - - -

**Input:**
```
theorem amc12a_2003_p24:
  fixes a b::real
  assumes "b \<le>a" and "1<b"
  shows "ln (a/b) / ln a + ln (b/a) / ln b \<le>0" (is "?L \<le> _")
proof -
  (* Step 1: Define x = ln a and y = ln b. *)
  define x y where "x=ln a" and "y=ln b"
  (* Step 2: With the conditions set as y > 0 and x ≥ y, we transition the equation to be framed in terms of x and y, thereby yielding
  (x−y)/x + (y−x)/y. *)

  (* Step 3: Simplify the expression to 2 − y/x − x/y. *)
  also have "... = 2 - (y/x + x/y)"
    using \<open>x>0\<close> \<open>y>0\<close> sledgehammer
  (* Step 4: Using the inequality ( √(x/y) − √(y/x) )² ≥ 0, it is shown that the simplified equation 2 − y/x − x/y is less than or equal to 0. *)
  also have "... \<le> 0"
  proof -
    have "sqrt (x/y) * sqrt (y/x) = 1"
      sledgehammer
    have "0\<le> (sqrt (x/y) - sqrt (y/x))^2"
      sledgehammer
    also have "... = x / y - 2 + y / x"
      unfolding power2_eq_square using \<open>x>0\<close> \<open>y>0\<close> \<open>sqrt (x/y) * sqrt (y/x) = 1\<close>
      sledgehammer
    finally show ?thesis
      sledgehammer
  qed
  finally show ?thesis.
qed
```

- - - - - - - - - - - - - - - - - - - - - - - - - - - - - - - - - - - - - - - - - - - - - - - - - - - - - - - - - - - - - - -

**Output:**
```
theorem amc12a_2003_p24:
  fixes a b::real
  assumes "b \<le>a" and "1<b"
  shows "ln (a/b) / ln a + ln (b/a) / ln b \<le>0" (is "?L \<le> _")
proof -
  (* Step 1: Define x = ln a and y = ln b. *)
  define x y where "x=ln a" and "y=ln b"
  (* Step 2: With the conditions set as y > 0 and x ≥ y, we transition the equation to be framed in terms of x and y, thereby yielding
  (x−y)/x + (y−x)/y. *)
  have "y>0"
    using \<open>b>1\<close> unfolding y_def using ln_gt_zero sledgehammer
  moreover have "x \<ge> y"
    using \<open>b\<le>a\<close> unfolding x_def y_def using ln_mono sledgehammer
  hence "(x - y)/x + (y - x)/y = 2 - y/x - x/y"
    using \<open>x>0\<close> \<open>y>0\<close> by (simp add: diff_divide_distrib)
  (* Step 3: Simplify the expression to 2 − y/x − x/y. *)
  also have "... = 2 - (y/x + x/y)"
    using \<open>x>0\<close> \<open>y>0\<close> sledgehammer
  (* Step 4: Using the inequality ( √(x/y) − √(y/x) )² ≥ 0, it is shown that the simplified equation 2 − y/x − x/y is less than or equal to 0. *)
  also have "... \<le> 0"
  proof -
    have "sqrt (x/y) * sqrt (y/x) = 1"
      sledgehammer
    have "0\<le> (sqrt (x/y) - sqrt (y/x))^2"
      sledgehammer
    also have "... = x / y - 2 + y / x"
      unfolding power2_eq_square using \<open>x>0\<close> \<open>y>0\<close> \<open>sqrt (x/y) * sqrt (y/x) = 1\<close>
      sledgehammer
    finally show ?thesis
      sledgehammer
  qed
  finally show ?thesis.
qed
```

Figure 8: An instance of the "verify" component within the Verify_and_correct function in Algorithm 2. ChatGPT encounters a *failure* in reconstructing the proof associated with *step 2*, thereby deeming it an unsuitable subgoal.

Your task is to complete the missing part of a Isabelle proof based on the corresponding description provided. The proof will consist of several steps, and you will need to use your knowledge of mathematical concepts and logical reasoning to fill in the gap.

**Input:**
```
theorem amc12a_2003_p24:
    fixes a b::real
    assumes "b \<le>a" and "1<b"
    shows "ln (a/b) / ln a + ln (b/a) / ln b \<le>0" (is "?L \<le> _")
proof –
    (* Step 1: Define x = ln a and y = ln b. *)
    define x y where "x=ln a" and "y=ln b"
    (* Step 2: With the conditions set as y > 0 and x ≥ y, we transition the equation to be framed in terms of x and y, thereby yielding
```
$\frac{x-y}{x} + \frac{y-x}{y}$. *)
```
    have "y>0"
        using \<open>b>1\<close> unfolding y_def using ln_gt_zero sledgehammer
    moreover have "x \<ge>y"
        using \<open>a\<ge>b\<close> unfolding x_def y_def using assms(2) sledgehammer
    ultimately have "x>0"
        sledgehammer
    have "?L = (x-y)/x + (y-x)/y"
        apply (subst (1 2) ln_div)
        using assms unfolding x_def y_def sledgehammer
    (* Step 3: Simplify the expression to 2 − y/x − x/y. *)
```
    (* Step 4: Using the inequality $(\sqrt{\frac{x}{y}} - \sqrt{\frac{y}{x}})^2 \geq 0$, it is shown that the simplified equation $2 - \frac{y}{x} - \frac{x}{y}$ is less than or equal to 0. *)
```
    also have "... \<le> 0"
    proof –
        have "sqrt (x/y) * sqrt (y/x) = 1"
            sledgehammer
        have "0\<le> (sqrt (x/y) - sqrt (y/x))^2"
            sledgehammer
        also have "... = x / y - 2 + y / x"
            unfolding power2_eq_square using \<open>x>0\<close> \<open>y>0\<close> \<open>sqrt (x/y) * sqrt (y/x) = 1\<close>
            sledgehammer
        finally show ?thesis
            sledgehammer
    qed
    finally show ?thesis.
qed
```

**Output:**
```
theorem amc12a_2003_p24:
    fixes a b::real
    assumes "b \<le>a" and "1<b"
    shows "ln (a/b) / ln a + ln (b/a) / ln b \<le>0" (is "?L \<le> _")
proof –
    (* Step 1: Define x = ln a and y = ln b. *)
    define x y where "x=ln a" and "y=ln b"
    (* Step 2: With the conditions set as y > 0 and x ≥ y, we transition the equation to be framed in terms of x and y, thereby yielding
```
$\frac{x-y}{x} + \frac{y-x}{y}$. *)
```
    have "y>0"
        using \<open>b>1\<close> unfolding y_def using ln_gt_zero sledgehammer
    moreover have "x \<ge>y"
        using \<open>a\<ge>b\<close> unfolding x_def y_def using assms(2) sledgehammer
    ultimately have "x>0"
        sledgehammer
    have "?L = (x-y)/x + (y-x)/y"
        apply (subst (1 2) ln_div)
        using assms unfolding x_def y_def sledgehammer
    (* Step 3: Simplify the expression to 2 − y/x − x/y. *)
    also have "... = 2 - (y/x + x/y)"
        using \<open>x>0\<close> \<open>y>0\<close> sledgehammer
```
    (* Step 4: Using the inequality $(\sqrt{\frac{x}{y}} - \sqrt{\frac{y}{x}})^2 \geq 0$, it is shown that the simplified equation $2 - \frac{y}{x} - \frac{x}{y}$ is less than or equal to 0. *)
```
    also have "... \<le> 0"
    proof –
        have "sqrt (x/y) * sqrt (y/x) = 1"
            sledgehammer
        have "0\<le> (sqrt (x/y) - sqrt (y/x))^2"
            sledgehammer
        also have "... = x / y - 2 + y / x"
            unfolding power2_eq_square using \<open>x>0\<close> \<open>y>0\<close> \<open>sqrt (x/y) * sqrt (y/x) = 1\<close>
            sledgehammer
        finally show ?thesis
            sledgehammer
    qed
    finally show ?thesis.
qed
```

Figure 9: An instance of the "verify" component within the Verify_and_correct function in Algorithm 2. ChatGPT *successfully* reconstructs the proof associated with *step 3*, thus validating it as a viable subgoal.

The Isabelle Proof Assistant was unable to verify Step 2. To resolve this, consider providing more detailed information or clarity to enhance the explanation for this particular step.

- - - - - - - - - - - - - - - - - - - - - - - - - - - - - - - - - - - - - - - - - - - - - - - - - - - - - - -

**Input:**
```
theorem amc12a_2003_p24:
    fixes a b::real
    assumes "b \<le>a" and "1<b"
    shows "ln (a/b) / ln a + ln (b/a) / ln b \<le>0" (is "?L \<le> _")
proof –
    (* Step 1: Define x = ln a and y = ln b. *)
    define x y where "x=ln a" and "y=ln b"
    (* Step 2: With the conditions set as y > 0 and x ≥ y, we transition the equation to be framed in terms of x and y, thereby yielding
    (x−y)/x + (y−x)/y. *)
    have "y>0"
        using \<open>b>1\<close> unfolding y_def using ln_gt_zero sledgehammer
    moreover have "x \<ge>y"
        using \<open>a\<ge>b\<close> unfolding x_def y_def using assms(2) sledgehammer
    ultimately have "x>0"
        sledgehammer
    have "?L = (x-y)/x + (y-x)/y"
        apply (subst (1 2) ln_div)
        using assms unfolding x_def y_def sledgehammer
    (* Step 3: Simplify the expression to 2 − y/x − x/y. *)
    also have "... = 2 - (y/x + x/y)"
        using \<open>x>0\<close> \<open>y>0\<close> sledgehammer
    (* Step 4: Using the inequality (√(x/y) − √(y/x))^2 ≥ 0, it is shown that the simplified equation 2 − y/x − x/y is less than or equal to 0. *)
    also have "... \<le> 0"
    proof –
        have "sqrt (x/y) * sqrt (y/x) = 1"
            sledgehammer
        have "0\<le> (sqrt (x/y) - sqrt (y/x))^2"
            sledgehammer
        also have "... = x / y - 2 + y / x"
            unfolding power2_eq_square using \<open>x>0\<close> \<open>y>0\<close> \<open>sqrt (x/y) * sqrt (y/x) = 1\<close>
            sledgehammer
        finally show ?thesis
            sledgehammer
    qed
    finally show ?thesis.
qed
```

- - - - - - - - - - - - - - - - - - - - - - - - - - - - - - - - - - - - - - - - - - - - - - - - - - - - - - -

**Output:**
```
theorem amc12a_2003_p24:
    fixes a b::real
    assumes "b \<le>a" and "1<b"
    shows "ln (a/b) / ln a + ln (b/a) / ln b \<le>0" (is "?L \<le> _")
proof –
    (* Step 1: Define x = ln a and y = ln b. *)
    define x y where "x=ln a" and "y=ln b"
    (* Step 2: Show that y > 0 and x ≥ y. *)
    have "y>0"
        using \<open>b>1\<close> unfolding y_def using ln_gt_zero sledgehammer
    moreover have "x \<ge>y"
        using \<open>a\<ge>b\<close> unfolding x_def y_def using assms(2) sledgehammer
    ultimately have "x>0"
        sledgehammer
    (* Step 3: Use the definitions of x and y to rewrite the expression as (x−y)/x + (y−x)/y. *)
    have "?L = (x-y)/x + (y-x)/y"
        apply (subst (1 2) ln_div)
        using assms unfolding x_def y_def sledgehammer
    (* Step 4: Simplify the expression to 2 − y/x − x/y. *)
    also have "... = 2 - (y/x + x/y)"
        using \<open>x>0\<close> \<open>y>0\<close> sledgehammer
    (* Step 5: Using the inequality (√(x/y) − √(y/x))^2 ≥ 0, it is shown that the simplified equation 2 − y/x − x/y is less than or equal to 0. *)
    also have "... \<le> 0"
    proof –
        have "sqrt (x/y) * sqrt (y/x) = 1"
            sledgehammer
        have "0\<le> (sqrt (x/y) - sqrt (y/x))^2"
            sledgehammer
        also have "... = x / y - 2 + y / x"
            unfolding power2_eq_square using \<open>x>0\<close> \<open>y>0\<close> \<open>sqrt (x/y) * sqrt (y/x) = 1\<close>
            sledgehammer
        finally show ?thesis
            sledgehammer
    qed
    finally show ?thesis.
qed
```

Figure 10: An instance of the "correct" component within the Verify_and_correct function in Algorithm 2. ChatGPT works on the decomposition of the original subgoal (specifically, step 2 in the input) into a series of more granular subgoals (specifically, step 2 - 3 in the output).

Your task is to complete the missing part of a Isabelle proof based on the corresponding description provided. The proof will consist of several steps, and you will need to use your knowledge of mathematical concepts and logical reasoning to fill in the gap.

**Input:**
```
theorem amc12a_2003_p24:
  fixes a b::real
  assumes "b \<le>a" and "1<b"
  shows "ln (a/b) / ln a + ln (b/a) / ln b \<le>0" (is "?L \<le> _")
proof -
  (* Step 1: Define x = ln a and y = ln b. *)
  define x y where "x=ln a" and "y=ln b"
  (* Step 2: Show that y > 0 and x ≥ y. *)

  (* Step 3: Use the definitions of x and y to rewrite the expression as x-y/x + y-x/y. *)
  have "?L = (x-y)/x + (y-x)/y"
    apply (subst (1 2) ln_div)
    using assms unfolding x_def y_def sledgehammer
  (* Step 4: Simplify the expression to 2 - y/x - x/y. *)
  also have "... = 2 - (y/x + x/y)"
    using \<open>x>0\<close> \<open>y>0\<close> sledgehammer
  (* Step 5: Using the inequality ( sqrt(x/y) - sqrt(y/x) )^2 ≥ 0, it is shown that the simplified equation 2 - y/x - x/y is less than or equal to 0. *)
  also have "... \<le> 0"
  proof -
    have "sqrt (x/y) * sqrt (y/x) = 1"
      sledgehammer
    have "0\<le> (sqrt (x/y) - sqrt (y/x))^2"
      sledgehammer
    also have "... = x / y - 2 + y / x"
      unfolding power2_eq_square using \<open>x>0\<close> \<open>y>0\<close> \<open>sqrt (x/y) * sqrt (y/x) = 1\<close>
      sledgehammer
    finally show ?thesis
      sledgehammer
  qed
  finally show ?thesis.
qed
```

**Output:**
```
theorem amc12a_2003_p24:
  fixes a b::real
  assumes "b \<le>a" and "1<b"
  shows "ln (a/b) / ln a + ln (b/a) / ln b \<le>0" (is "?L \<le> _")
proof -
  (* Step 1: Define x = ln a and y = ln b. *)
  define x y where "x=ln a" and "y=ln b"
  (* Step 2: Show that y > 0 and x ≥ y. *)
  have "y>0"
    using \<open>b>1\<close> unfolding y_def using ln_gt_zero sledgehammer
  moreover have "x \<ge>y"
    using \<open>a\<ge>b\<close> unfolding x_def y_def using assms(2) sledgehammer
  ultimately have "x>0"
    sledgehammer
  (* Step 3: Use the definitions of x and y to rewrite the expression as x-y/x + y-x/y. *)
  have "?L = (x-y)/x + (y-x)/y"
    apply (subst (1 2) ln_div)
    using assms unfolding x_def y_def sledgehammer
  (* Step 4: Simplify the expression to 2 - y/x - x/y. *)
  also have "... = 2 - (y/x + x/y)"
    using \<open>x>0\<close> \<open>y>0\<close> sledgehammer
  (* Step 5: Using the inequality ( sqrt(x/y) - sqrt(y/x) )^2 ≥ 0, it is shown that the simplified equation 2 - y/x - x/y is less than or equal to 0. *)
  also have "... \<le> 0"
  proof -
    have "sqrt (x/y) * sqrt (y/x) = 1"
      sledgehammer
    have "0\<le> (sqrt (x/y) - sqrt (y/x))^2"
      sledgehammer
    also have "... = x / y - 2 + y / x"
      unfolding power2_eq_square using \<open>x>0\<close> \<open>y>0\<close> \<open>sqrt (x/y) * sqrt (y/x) = 1\<close>
      sledgehammer
    finally show ?thesis
      sledgehammer
  qed
  finally show ?thesis.
qed
```

Figure 11: An instance of the "verify" component within the Verify_and_correct function in Algorithm 2. ChatGPT *successfully* reconstructs the proof associated with *step 2*, thus validating it as a viable subgoal.

Your task is to complete the missing part of a Isabelle proof based on the corresponding description provided. The proof will consist of several steps, and you will need to use your knowledge of mathematical concepts and logical reasoning to fill in the gap.

- - - - - - - - - - - - - - - - - - - - - - - - - - - - - - - - - - - - - - - - - - - - - - - - - - - - - - - - - - - - -

**Input:**
```
theorem amc12a_2003_p24:
    fixes a b::real
    assumes "b \<le>a" and "1<b"
    shows "ln (a/b) / ln a + ln (b/a) / ln b \<le>0" (is "?L \<le> _")
proof −
    (* Step 1: Define x = ln a and y = ln b. *)
    define x y where "x=ln a" and "y=ln b"
    (* Step 2: Show that y > 0 and x ≥ y. *)
    have "y>0"
        using \<open>b>1\<close> unfolding y_def using ln_gt_zero sledgehammer
    moreover have "x \<ge>y"
        using \<open>a\<ge>b\<close> unfolding x_def y_def using assms(2) sledgehammer
    ultimately have "x>0"
        sledgehammer
    (* Step 3: Use the definitions of x and y to rewrite the expression as $\frac{x-y}{x} + \frac{y-x}{y}$. *)

    (* Step 4: Simplify the expression to $2 - \frac{y}{x} - \frac{x}{y}$. *)
    also have "... = 2 - (y/x + x/y)"
        using \<open>x>0\<close> \<open>y>0\<close> sledgehammer
    (* Step 5: Using the inequality $(\sqrt{\frac{x}{y}} - \sqrt{\frac{y}{x}})^2 \geq 0$, it is shown that the simplified equation $2 - \frac{y}{x} - \frac{x}{y}$ is less than or equal to 0. *)
    also have "... \<le> 0"
        proof −
        have "sqrt (x/y) * sqrt (y/x) = 1"
            sledgehammer
        have "0\<le> (sqrt (x/y) - sqrt (y/x))^2"
            sledgehammer
        also have "... = x / y - 2 + y / x"
            unfolding power2_eq_square using \<open>x>0\<close> \<open>y>0\<close> \<open>sqrt (x/y) * sqrt (y/x) = 1\<close>
            sledgehammer
        finally show ?thesis
            sledgehammer
        qed
    finally show ?thesis.
qed
```

- - - - - - - - - - - - - - - - - - - - - - - - - - - - - - - - - - - - - - - - - - - - - - - - - - - - - - - - - - - - -

**Output:**
```
theorem amc12a_2003_p24:
    fixes a b::real
    assumes "b \<le>a" and "1<b"
    shows "ln (a/b) / ln a + ln (b/a) / ln b \<le>0" (is "?L \<le> _")
proof −
    (* Step 1: Define x = ln a and y = ln b. *)
    define x y where "x=ln a" and "y=ln b"
    (* Step 2: Show that y > 0 and x ≥ y. *)
    have "y>0"
        using \<open>b>1\<close> unfolding y_def using ln_gt_zero sledgehammer
    moreover have "x \<ge>y"
        using \<open>a\<ge>b\<close> unfolding x_def y_def using assms(2) sledgehammer
    ultimately have "x>0"
        sledgehammer
    (* Step 3: Use the definitions of x and y to rewrite the expression as $\frac{x-y}{x} + \frac{y-x}{y}$. *)
    have "?L = (x-y)/x + (y-x)/y"
        apply (subst (1 2) ln_div)
        using assms unfolding x_def y_def sledgehammer
    (* Step 4: Simplify the expression to $2 - \frac{y}{x} - \frac{x}{y}$. *)
    also have "... = 2 - (y/x + x/y)"
        using \<open>x>0\<close> \<open>y>0\<close> sledgehammer
    (* Step 5: Using the inequality $(\sqrt{\frac{x}{y}} - \sqrt{\frac{y}{x}})^2 \geq 0$, it is shown that the simplified equation $2 - \frac{y}{x} - \frac{x}{y}$ is less than or equal to 0. *)
    also have "... \<le> 0"
        proof −
        have "sqrt (x/y) * sqrt (y/x) = 1"
            sledgehammer
        have "0\<le> (sqrt (x/y) - sqrt (y/x))^2"
            sledgehammer
        also have "... = x / y - 2 + y / x"
            unfolding power2_eq_square using \<open>x>0\<close> \<open>y>0\<close> \<open>sqrt (x/y) * sqrt (y/x) = 1\<close>
            sledgehammer
        finally show ?thesis
            sledgehammer
        qed
    finally show ?thesis.
qed
```

Figure 12: An instance of the "verify" component within the Verify_and_correct function in Algorithm 2. ChatGPT *successfully* reconstructs the proof associated with *step 3*, thus validating it as a viable subgoal.

## B    MORE DETAILS ABOUT DEMONSTRATION REORGANIZATION

### B.1    PARAMETERIZATION

In alignment with Austin et al. (2021), we adopt discrete diffusion models to model binary random variables. Explicitly, the forward process is given by:

$$q(\boldsymbol{\psi}_t|\boldsymbol{\psi}_{t-1}) = \mathrm{Cat}\left(\boldsymbol{\psi}_t; \mathbf{p} = \delta(\boldsymbol{\psi}_{t-1})\boldsymbol{Q}_t\right), \tag{5}$$

where $\delta(\boldsymbol{\psi})$ symbolizes the one-hot encoding of $\boldsymbol{\psi}$, $\boldsymbol{Q}_t = \begin{bmatrix} (1-\beta_t) & \beta_t \\ \beta_t & (1-\beta_t) \end{bmatrix}$ denotes the transition matrix, $\beta_t$ corresponds to the corruption ratio and satisfies that $\prod_{t=1}^{T}(1-\beta_t) \approx 0$. The marginal at step $t$ and the posterior at step $t-1$ can be articulated as:

$$q(\boldsymbol{\psi}_t|\boldsymbol{\psi}_0) = \mathrm{Cat}\left(\boldsymbol{\psi}_t; \mathbf{p} = \delta(\boldsymbol{\psi}_0)\overline{\boldsymbol{Q}}_t\right),$$

$$q(\boldsymbol{\psi}_{t-1}|\boldsymbol{\psi}_t, \boldsymbol{\psi}_0) = \mathrm{Cat}\left(\boldsymbol{\psi}_{t-1}; \mathbf{p} = \frac{\delta(\boldsymbol{\psi}_t)\boldsymbol{Q}_t^\top \odot \delta(\boldsymbol{\psi}_0)\overline{\boldsymbol{Q}}_{t-1}}{\delta(\boldsymbol{\psi}_0)\overline{\boldsymbol{Q}}_t\delta(\boldsymbol{\psi}_t)^\top}\right), \tag{6}$$

where $\overline{\boldsymbol{Q}}_t = \boldsymbol{Q}_1\boldsymbol{Q}_2\ldots\boldsymbol{Q}_t$. In consonance with Austin et al. (2021), we employ a denoising neural network which is tasked with the prediction of $p(\boldsymbol{\psi_0}|\boldsymbol{\psi_t})$, thereby enabling the parameterization of the reverse process:

$$p_{\boldsymbol{\theta}}(\boldsymbol{\psi}_{t-1}|\boldsymbol{\psi}_t, x) \propto \sum_{\boldsymbol{\psi}} q(\boldsymbol{\psi}_{t-1}|\boldsymbol{\psi}_t, \boldsymbol{\psi}_0)p_{\boldsymbol{\theta}}(\boldsymbol{\psi}_0|\boldsymbol{\psi}_t, x). \tag{7}$$

### B.2    IMPLEMENTATION OF GNN

Our work employs a modified version of GNN, a model that exhibits anisotropic characteristics and is enhanced by edge gating methodologies (Bresson & Laurent, 2018; Sun & Yang, 2023). We define $\mathbf{t}$ as sinusoidal representations (Vaswani et al., 2017) associated with the denoising timestep $t$. Consider $\boldsymbol{h}_i^\ell$ and $\boldsymbol{e}_{ij}^\ell$ as the features of node $i$ and edge $ij$ at a specific layer $\ell$, respectively. During the transition between layers, these features disseminate via an anisotropic message propagation paradigm as follows:

$$\hat{\boldsymbol{e}}_{ij}^{\ell+1} = \boldsymbol{P}^\ell \boldsymbol{e}_{ij}^\ell + \boldsymbol{Q}^\ell \boldsymbol{h}_i^\ell + \boldsymbol{R}^\ell \boldsymbol{h}_j^\ell,$$

$$\boldsymbol{e}_{ij}^{\ell+1} = \boldsymbol{e}_{ij}^\ell + \mathrm{MLP}_e(\mathrm{BN}(\hat{\boldsymbol{e}}_{ij}^{\ell+1})) + \mathrm{MLP}_t(\mathbf{t}), \tag{8}$$

$$\boldsymbol{h}_i^{\ell+1} = \boldsymbol{h}_i^\ell + \mathrm{ReLU}(\mathrm{BN}(\boldsymbol{U}^\ell \boldsymbol{h}_i^\ell + \mathrm{SUM}_{j \in \mathcal{N}_i}(\sigma(\hat{\boldsymbol{e}}_{ij}^{\ell+1}) \odot \boldsymbol{V}^\ell \boldsymbol{h}_j^\ell))),$$

where $\boldsymbol{P}^\ell, \boldsymbol{Q}^\ell, \boldsymbol{R}^\ell, \boldsymbol{U}^\ell, \boldsymbol{V}^\ell \in \mathbb{R}^{d \times d}$ denote layer-specific learnable parameters with $d$ denoting the dimension of hidden state. BN signifies the Batch Normalization operation (Ioffe & Szegedy, 2015), while SUM represents sum pooling. $\odot$ designates the Hadamard product, and $\mathcal{N}_i$ encapsulates the set of neighboring nodes of node $i$. Lastly, a two-layer multi-layer perceptron is denoted by $\mathrm{MLP}(\cdot)$.

In our experiments, we define $\boldsymbol{h}_i^0 = \boldsymbol{W}[\mathrm{Ada}(x); \mathrm{Ada}(E_i^{(K)})]$ where $\boldsymbol{W} \in \mathbb{R}^{d \times 3072}$ is a learnable parameter. $\mathrm{Ada}(x), \mathrm{Ada}(E_i^{(K)}) \in \mathbb{R}^{1536 \times 1}$ denote the ada embeddings [6] of the statement $x$ and the $i$-th demonstration example, respectively. The operator $[\cdot; \cdot]$ denotes the concatenation operation between two vectors. $\boldsymbol{e}_{ij}^0$ are initialized as sinusoidal features of the edges.

### B.3    SAMPLING PROCESS

A straightforward strategy for creating a demonstration organization is by directly sampling $\boldsymbol{\psi} \sim p_{\boldsymbol{\theta}}(\boldsymbol{\psi}_0|x)$. However, this strategy introduces two key challenges: (1) A cycle in $\boldsymbol{\psi}$ might be present, indicating that at least one demonstration example is selected multiple times; (2) $\boldsymbol{\psi}$ could include multiple separate sub-graphs, making it difficult to define the relative position between two demonstration examples from two different sub-graphs. Taking a cue from treating diffusion models as discriminative approaches (Li et al., 2023), we start by randomly creating 200 potential

---

[6]https://platform.openai.com/docs/guides/embeddings

solutions. Using the diffusion model's ability to provide conditional density estimates, we rate these 200 potential solutions and select the one with the highest score to build the final demonstration organization. We then reconstruct the sequence of demonstration examples from $\psi$, adding examples one by one into the LLM context until we hit the length limit of the LLM.

### B.4 HYPERPARAMETERS AND HARDWARE SETUP

In the course of our experiment, we employ a 3-layer Anisotropic Graph Neural Network with a hidden state dimensionality set to 256. We sweep the learning rate from $[1e-4, 2e-4, 5e-4, 7e-4]$ and sweep batch size from $[4, 8, 16, 32]$. The processes of training and inference for the diffusion models are conducted on a NVIDIA RTX 3090 GPU.

## C MORE IN-DEPTH ANALYSIS

### C.1 IMPACT OF PROMPT WORDING

To systematically investigate the influence of varying prompt wordings on the effectiveness of our proposed method, we further conduct an experiment on the miniF2F-test set. The specific prompt is as follows:

```
Assume the role of a mathematician proficient in Isabelle.
When provided with both informal and formal statements of a problem,
your responsibility is to formulate a formal proof that Isabelle
can verify.
```

| Prompt Type | pass@100 |
|---|---|
| Original (Proposed) | 45.5% |
| Alternative Prompt | 41.0% |

Table 3: Comparison of pass rates with different prompt wordings over 100 autoformalization attempts on the miniF2F-test set.

As can be observed in Table 3, the original prompt yielded slightly better results compared to the alternative prompt. This suggests that the phrasing of the prompt can have an influence on the performance, and our original choice was more effective in achieving higher pass rates.

### C.2 SENSITIVITY TO EXAMPLE ORDERING

To assess the robustness and adaptability of our model, we explored the sensitivity of the model to the ordering of examples. Specifically, we maintained the examples derived from our diffusion model but altered their sequence. The experiment involves randomizing the sequence of examples after they were organized using the diffusion model. We then evaluated the model's ability to solve problems under this modified setup. The evaluation was conducted over 20 autoformalization attempts on the miniF2F-test set.

| Method | # of Problems Solved |
|---|---|
| Ours (Ordered) | 94 |
| Shuffled Order | 39 |

Table 4: Impact of example ordering.

The results in Table 4 demonstrate that the ordering of examples significantly impacts the model's performance. It is evident that the structured order provided by the diffusion model is important, as disrupting this order resulted in a substantial decline in the number of problems solved.

## C.3 IMPACT OF DEMONSTRATION SELECTION AND PRESENTATION ORDER

We conducted experiments comparing our method, which meticulously selects and orders demonstrations, to an alternative approach where the demonstrations are shuffled.

| Method | # of Problems Solved |
|---|---|
| Ours | 94 |
| Shuffled | 39 |

Table 5: Comparison of our method vs. shuffled demonstrations

The results in Table 5 underscore the significance of both the selection and the order in which demonstrations are presented. It is evident that a methodical approach to selecting and ordering demonstrations yields superior results compared to a scenario where demonstrations are presented in a shuffled manner.

## C.4 IMPACT OF INCREASING DEMONSTRATIONS

We conduct experiments to elucidate the impact of varying the number of demonstrations on the performance of our model.

| Number of Demonstrations | # of Problems Solved |
|---|---|
| Ours | 94 |
| 1 example | 16 |
| 2 examples | 29 |
| 3 examples | 34 |

Table 6: Performance analysis with varying numbers of demonstrations.

The empirical results in Table 6 underscore the collective significance of all demonstrations in enhancing the model's performance. It is evident that the incorporation of multiple demonstrations is pivotal, substantiating the efficacy of our diffusion-based model.

## C.5 IMPACT OF RANDOMLY GENERATED PROBLEM NAMES

The incorporation of original problem names could potentially facilitate models like ChatGPT in associating them to readily available solutions online. This scenario posits a risk of primarily assessing the model's capability to translate human proof sketches to subgoals and formal proofs, rather than its proficiency in innovating novel subgoals. To mitigate this concern and uphold the rigor of our evaluation, we initiated experiments substituting original problem names with random identifiers and analyzed the ensuing impact on performance. The evaluation was conducted over 20 autoformalization attempts on the miniF2F-test set.

| Method | Original | Random |
|---|---|---|
| Ours | 38.5% | 39.8% |
| Top-K | 35.7% | 34.8% |

Table 7: Performance comparison with original and random problem names on the miniF2F-test set.

The results in Table 7 indicate that our method maintains robust performance, even manifesting a marginal enhancement when random names are deployed. This steadfastness in performance, regardless of the naming conventions adopted, accentuates the robustness and adaptability of our approach across diverse experimental conditions.

## C.6 IMPACT OF PROOF SKETCH QUALITY

To comprehensively assess the robustness of our method in relation to the quality of informal proof sketches, we conducted an additional experiment that involved the generation of informal proofs using gpt-3.5-turbo-0613 for problems from the miniF2F-valid dataset. Utilizing the methodology outlined in §2.1, we constructed subgoal-based proofs for these informal proofs. For this purpose, a set of 61 problems was carefully chosen to serve as demonstration samples, aligning with the experimental setup specified in §3.4. The results of this experiment are shown in Table 8.

| Method | valid | test |
|---|---|---|
| Ours (DSP Informal Proofs) | 48.0% (±0.4) | 45.5% (±0.6) |
| Ours (GPT-3.5 Informal Proofs) | 47.7% (±0.5) | 45.0% (±0.7) |

Table 8: Performance comparison with varying proof sketch quality.

The results from Table 8 demonstrate that our method maintains a notable degree of robustness against variations in the quality of informal proof sketches. This robustness indicates the adaptability and dependability of our approach when confronted with different levels of proof sketch quality, affirming its practical applicability in diverse scenarios.

## C.7 EVALUATION OF OPTIMAL ORGANIZATION SEARCH

To further evaluate the efficiency of our method, we conducted experiments involving 200 randomized organization sampling attempts for each statement within the miniF2F-valid and miniF2F-test datasets. These efforts aimed to determine the potential of finding an optimal organization for each statement. The results of this extensive search are displayed in Table 9.

| Method | valid | test |
|---|---|---|
| Ours | 48.0% | 45.5% |
| Optimal Organization Search | 48.4% | 44.3% |

Table 9: Performance comparison with optimal organization search.

These results corroborate previous observations (Jiang et al., 2022a) that the advantages of extensive organization searches tend to level off beyond a certain number of attempts (in this case, 100). These results highlight the inherent challenges of the miniF2F dataset and emphasize the effectiveness of our diffusion model.

## C.8 ANALYSIS OF SLEDGEHAMMER USAGE

In order to provide a comprehensive understanding of how our method and DSP (Jiang et al., 2022a) utilize the Sledgehammer tool, we have conducted a detailed analysis. This analysis was carried out on a machine with 64 CPU cores, focusing on the average number of Sledgehammer calls and their execution times for each solved statement. The results are shown in the Table 10, which illustrates the average number of Sledgehammer calls and their corresponding durations (in seconds) for each method.

| Method | Calls | | Duration (seconds) | |
|---|---|---|---|---|
| | valid | test | valid | test |
| DSP | 2.33 | 2.39 | 3.29 | 2.98 |
| Ours | 2.88 | 3.22 | 4.16 | 4.94 |

Table 10: Average number of Sledgehammer calls and durations.

The results in Table 10 indicate that our method exhibits a slight increase in both the frequency of Sledgehammer calls and their execution times in comparison with DSP. Specifically, this increase is

primarily observed in statements that our method can solve but DSP cannot. For these statements, the number of Sledgehammer calls on the miniF2F-valid and miniF2F-test are 3.21 and 5.38, respectively. This suggests that the need for Sledgehammer becomes more important as the problem complexity increases.

## D    DISCUSSIONS ABOUT CORRELATION BETWEEN INPUT STATEMENTS AND ORGANIZATION

We further investigate the relationship between input statements and the demonstration organization generated by diffusion models within the miniF2F dataset. Our aim was to determine whether the diffusion model tends to generate a generic organization broadly applicable across various statements or if it tailors unique organizations to individual statements.

The empirical results suggest that there is not a "one-size-fits-all" optimal demonstration organization. Specifically, even the most adaptable organization we identified could successfully prove only **3** distinct statements. Furthermore, we only observed **4** instances of such adaptable organizations on the miniF2F-test. This highlights the tailored nature of each organization to its corresponding statement. The limited scope of even the most adaptable organization indicates the challenge in identifying the optimal demonstration organization for each unique statement. This complexity further emphasizes the effectiveness of our model in efficiently searching for and determining the most suitable demonstration organization for individual statements.

## E    DISCUSSIONS ABOUT THE TRAIN/TEST DATA LEAKAGE

The potential for train/test leakage, stemming from the use of original problem names from the miniF2F dataset, was brought to our attention. In response, we undertook a thorough analysis. Our findings indicated that about 32.0% of problems could be identifiable by ChatGPT based on their names, and approximately 40% of informal statements were susceptible to leakage. To further elucidate this, a comprehensive manual search was conducted on the test set's informal statements and their corresponding proofs. This search revealed that around 40% had been available online before September 1, 2021, with platforms like Mathway and Mathematics Stack Exchange being significant contributors. Despite a number of miniF2F problems lacking Isabelle solutions, we revised the potentially leaked statements. Experiments with randomized identifiers further confirmed that our method's performance is not contingent on data leakage (see Appendix C.5).

## F    ADDITIONAL EXAMPLES

We provide additional cases in Figure 13 to 18 to demonstrate the efficacy of our method.[7]

Figures 13 to 15 display examples where our method effectively guides the proof process. For instance, in Figure 13, the method leverages demonstrations involving division and modulus operations, successfully navigating towards a clear proof path. Similarly, in Figure 14, it utilizes demonstrations based on squaring and square root operations, accurately predicting potential subgoals. Figure 15 continues this trend, demonstrating our method's consistent ability to discern viable subgoals, thus facilitating the construction of proofs.

Additionally, Figures 16 to 18 display the same statements as Figures 13 to 15 but with the demonstration examples reordered. This reorganization illustrates the impact of demonstration order on the generation of proofs, revealing how changes in the order can lead to unexpected errors. The comparison between these two groups highlights the significance of demonstration organization in influencing final performance.

---

[7]In Figures 13 to 18, identical statements are guided by a consistent set of demonstration examples, but in a different order. Some less relevant demonstration examples have been omitted due to space constraints.

As a mathematician familiar with Isabelle, your task is to provide a formal proof in response to a given problem statement.
Your proof should be structured and clearly written, meeting the following criteria:
- It can be verified by Isabelle.
- Each step of the proof should be explained in detail using comments enclosed in "(*" and "*)".
- The explanation for each step should be clear and concise, avoiding any unnecessary or apologetic language.
Please ensure that your proof is well-organized and easy to follow, with each step building upon the previous one.

**Problem:**
Let $n$ be an integer and $p$ be a prime number. Show that $p$ divides $n$ if and only if $n^2$ is congruent to 0 modulo $p$.

**Proof:**
```
theorem numbertheory_prmdvsneqnsqmodpeq0:
  fixes n :: int
    and p :: nat
  assumes "prime p"
  shows "p dvd n \<longleftrightarrow> (n^2) mod p = 0"
  (* Step 1: Assume that p divides n. Show that n² is congruent to 0 modulo p.
     Step 2: Assume that n² is congruent to 0 modulo p. Show that p divides n. *)
proof -
  (* Step 1: Assume that p divides n. Show that n² is congruent to 0 modulo p.*)
  assume "p dvd n"
  then show "(n^2) mod p = 0" sledgehammer
  (* Step 2: Assume that n² is congruent to 0 modulo p. Show that p divides n. *)
  next assume "(n^2) mod p = 0"
  then show "p dvd n" sledgehammer
qed
```

**Problem:**
Prove that for any natural number $n$, 3 divides $n^3 + 2n$.

**Proof:**
```
theorem induction_divisibility_3divnto3m2n:
  fixes n::nat
  shows "3 dvd n^3 + 2 * n".
  (* Step 1: Prove the base case n = 0.
     Step 2: Assume the statement is true for some n and prove it for n + 1.
     Step 3: Expand (n + 1)³+2(n + 1) and simplify.
     Step 4: Use the assumption that 3 divides n³ + 2n to show that 3 divides 3n² + 3n + 3.
     Step 5: Combine steps 3 and 4 to show that 3 divides (n + 1)³+2(n + 1) .
     Step 6: Conclude that the statement is true for all natural numbers n. *)
proof (induct n)
  (* Step 1: Prove the base case n = 0.*)
  case 0
    then show ?case sledgehammer
  (* Step 2: Assume the statement is true for some n and prove it for n + 1.*)
  next case (Suc n)
  have ?case when "n=0"
    using that sledgehammer
  moreover have ?case when "n>0"
    proof -
  (* Step 3: Expand (n + 1)³+2(n + 1) and simplify. *)
    have "(n + 1)^3 + 2 * (n + 1) = (n^3+3*n^2+3*n+1) + (2*n+2)"
      unfolding power3_eq_cube power2_eq_square sledgehammer
    also have "... = n^3+2*n + 3*n^2+3*n+3" sledgehammer
    finally have "(n+1)^3+2*(n+1) = n^3+2*n + 3*n^2+3*n+3".
  (* Step 4: Use the assumption that 3 divides n³ + 2n to show that 3 divides 3n² + 3n + 3.*)
    moreover have "3 dvd n^3+2*n" using Suc(1) sledgehammer
    moreover have "3 dvd 3*n^2+3*n+3" sledgehammer
  (* Step 5: Combine steps 3 and 4 to show that 3 divides (n + 1)³+2(n + 1) .*)
    ultimately have "3 dvd (n+1)^3+2*(n+1)" sledgehammer
  (* Step 6: Conclude that the statement is true for all natural numbers n. *)
    then show ?thesis sledgehammer
  qed
  ultimately show ?case sledgehammer
qed
```

**Problem:**
Suppose $n$ is a natural number such that $n$ leaves a remainder of 3 when divided by 7. Show that $(2n + 1)$ leaves a remainder of 0 when divided by 7.

**Proof:**
```
theorem mathd_numbertheory_370:
  fixes n :: nat
  assumes h0 : "n mod 7 = (3::nat)"
  shows "(2*n+1) mod 7 = (0::nat)"
```

*Prompt*

--------------------------------------------------------------------------------

*ChatGPT Output*

```
  (* Step 1: Express n as 7k + 3 for some k∈ℕ$.
     Step 2: Substitute the expression for n into the expression for 2n + 1 and simplify.
     Step 3: Show that the resulting expression is divisible by 7. *)
proof -
  (* Step 1: Express n as 7k + 3 for some k∈ℕ$.*)
  have "n = 7 * (n div 7) + (n mod 7)" using div_mult_mod_eq[of "n" "7"] sledgehammer
  also have "... = 7 * (n div 7) + 3" using h0 sledgehammer
  finally have c0: "n = 7 * (n div 7) + 3".
  (* Step 2: Substitute the expression for n into the expression for 2n + 1 and simplify.*)
  have "2*n+1 = 2*(7*(n div 7) + 3) + 1" using c0 sledgehammer
  also have "... = 14*(n div 7) + 7" sledgehammer
  also have "... = 7*(2*(n div 7) + 1)" sledgehammer
  finally have c1: "2*n+1 = 7*(2*(n div 7) + 1)".
  (* Step 3: Show that the resulting expression is divisible by 7. *)
  show "(2*n+1) mod 7 = 0" using c1 sledgehammer
qed
```

Figure 13: A case from miniF2F-valid. Our method identifies viable subgoals and leverages demonstration examples with division and modulus operations to complete the proof.

As a mathematician familiar with Isabelle, your task is to provide a formal proof in response to a given problem statement.
Your proof should be structured and clearly written, meeting the following criteria:
- It can be verified by Isabelle.
- Each step of the proof should be explained in detail using comments enclosed in "(*" and "*)".
- The explanation for each step should be clear and concise, avoiding any unnecessary or apologetic language.
Please ensure that your proof is well-organized and easy to follow, with each step building upon the previous one.

**Problem:**

Suppose that $x$ is a positive real number. Show that $2 - \sqrt{2} \geq 2 - x - \frac{1}{2x}$.

**Proof:**
```
theorem algebra_amgm_faxinrrp2msqrt2geq2mxm1div2x:
  fixes x :: real
  assumes "x > 0"
  shows "2 - sqrt 2 \<ge> 2 - x - 1/ (2 * x)"
  (* Step 1: Show that √2 ≤ x + 1/2x.
     Step 2: Rearrange the inequality to get 2x² + 1 − 2x√2 ≥ 0.
     Step 3: Factor the expression to get (√2x − 1)² ≥ 0.
     Step 4: Conclude that the inequality is true. *)
proof -
  (* Step 1: Show that √2 ≤ x + 1/2x. *)
  have c0: "2 * x > 0" using assms sledgehammer
  have "sqrt 2 \<le> x + 1 / (2 * x)"
  proof -
    define y where "y = sqrt 2"
    have c1: "2 = y * y"
    proof -
      have "2 = (sqrt 2) * (sqrt 2)" sledgehammer
      then have "... = y * y" using \<open>y = sqrt 2\<close> sledgehammer
      then show ?thesis sledgehammer
    qed
    have "(2 * x) * x + 1 - (2 * x) * (sqrt 2) = (y * y * x * x) + 1 - (2 * x) * y" using c1 y_def sledgehammer
    also have "... = (y*x) * (y*x) - 2 * (y*x) + 1" sledgehammer
    also have "... = (y*x - 1) * (y*x - 1)" sledgehammer
    also have "... \<ge> 0" sledgehammer
    ultimately have c2: "(2 * x) * x + 1 - (2 * x) * (sqrt 2) \<ge> 0" sledgehammer
    have "(2*x) * (x + 1/(2*x) - sqrt 2) = (2 * x) * x + (2 * x) * (1/(2*x)) - (2*x) * sqrt 2" sledgehammer
    also have "... = (2 * x) * x + 1 - (2*x) * sqrt 2" using c0 sledgehammer
    (* Step 2: Rearrange the inequality to get 2x² + 1 − 2x√2 ≥ 0.*)
    also have "... \<ge> 0" using c2 sledgehammer
    (* Step 3: Factor the expression to get (√2x − 1)² ≥ 0.*)
    ultimately have "(2*x) * (x + 1/(2*x) - sqrt 2) \<ge> 0" sledgehammer
    hence "x + 1/(2*x) - sqrt 2 \<ge> 0" using mult_pos_pos sledgehammer
    (* Step 4: Conclude that the inequality is true. *)
    then show ?thesis sledgehammer
  qed
  then show ?thesis sledgehammer
qed
```

**Problem:**

Suppose that $x$ and $y$ are two real numbers such that $x + y = 13$ and $xy = 24$. Find $\sqrt{x^2 + y^2}$ and show that it is equal to 11.

**Proof:**
```
theorem mathd_algebra_510:
  fixes x y :: real
  assumes h0 : "x+y=13"
    and h1 : "x*y=24"
  shows "sqrt (x^2 + y^2) = 11"
  (* Step 1: Use the identity x² + y² = (x + y)² − 2xy to find x² + y² in terms of x and y.
     Step 2: Substitute the given values of x and y to find x² + y² .
     Step 3: Show that x² + y² = 121.
     Step 4: Take the square root of both sides to find √(x² + y²).
     Step 5: Show that √(x² + y²) = 11. *)
proof -
  (* Step 1: Use the identity x² + y² = (x + y)² − 2xy to find x² + y² in terms of x and y. *)
  have c0: "x^2 + y^2 = (x+y)^2 - 2*x*y" sledgehammer
  (* Step 2: Substitute the given values of x and y to find x² + y² . *)
  also have "... = (13)^2 - 2*24" using h0 h1 sledgehammer
  (* Step 3: Show that x² + y² = 121. *)
  also have "... = 121" sledgehammer
  finally have c1: "x^2 + y^2 = 121" sledgehammer
  (* Step 4: Take the square root of both sides to find √(x² + y²). *)
  then show ?thesis sledgehammer
  (* Step 5: Show that √(x² + y²) = 11. *)
qed
```

**Problem:**

Suppose that $x$ and $y$ are two real numbers such that $\frac{x+y}{2} = 7$ and $\sqrt{xy} = \sqrt{19}$. Find $x^2 + y^2$.

**Proof:**
```
theorem mathd_algebra_332:
  fixes x y ::real
  assumes "(x + y) / 2 = 7"
    and "sqrt (x * y) = sqrt 19"
  shows "x^2 + y^2 = 158"
```

*Prompt*

------------------------------------------------------------------------------------

*ChatGPT Output*

```
  (* Step 1: Use the identity (x + y)² = x² + 2xy + y² to find x² + y² in terms of x + y and xy.
     Step 2: Substitute the given values of x + y and xy to find x² + y².
     Step 3: Show that x² + y² = 158. *)
proof -
  (* Step 1: Use the identity (x + y)² = x² + 2xy + y² to find x² + y² in terms of x + y and xy. *)
  have c0: "x^2 + y^2 = (x+y)^2 - 2*x*y" sledgehammer
  (* Step 2: Substitute the given values of x + y and xy to find x² + y². *)
  also have "... = (2*7)^2 - 2*(sqrt 19)^2" using assms sledgehammer
  (* Step 3: Show that x² + y² = 158. *)
  also have "... = 158" sledgehammer
  finally show "x^2 + y^2 = 158" sledgehammer
qed
```

Figure 14: A case from miniF2F-test. Our method predicts viable subgoals and utilizes demonstration examples involving squaring and square root operations to finalize the proof.

As a mathematician familiar with Isabelle, your task is to provide a formal proof in response to a given problem statement.
Your proof should be structured and clearly written, meeting the following criteria:
- It can be verified by Isabelle.
- Each step of the proof should be explained in detail using comments enclosed in "(*" and "*)".
- The explanation for each step should be clear and concise, avoiding any unnecessary or apologetic language.
Please ensure that your proof is well-organized and easy to follow, with each step building upon the previous one.

**Problem:**

Given a real number $n$ such that $n \neq 3$ and $\frac{(n+5)}{(n-3)} = 2$, show that $n = 11$.

**Proof:**

```
theorem mathd_algebra_181:
   fixes n :: real
   assumes h0 : "n \<noteq> 3" and h1 : "(n+5) / (n-3) = 2"
   shows "n=11"
   (* Step 1: Use the given equation (n+5)/(n-3) = 2 to obtain an equation involving n.
      Step 2: Simplify the equation to obtain an expression for n.
      Step 3: Show that the expression for n is equal to 11. *)
proof -
   (* Step 1: Use the given equation (n+5)/(n-3) = 2 to obtain an equation involving n. *)
   have "n+5 = 2 * (n-3)" using h0 h1 sledgehammer
   (* Step 2: Simplify the equation to obtain an expression for n. *)
   thus ?thesis sledgehammer
   (* Step 3: Show that the expression for n is equal to 11. *)
qed
```

**Problem:**

Prove by induction that the sum of the first $n$ odd numbers is $n^2$.

**Proof:**

```
theorem induction_sum_odd:
   fixes n :: nat
   assumes "n > 0"
   shows "(\<Sum> (k::nat) = 0..(n-1). 2 * k + 1) = n^2"
   (* Step 1: Base case: Show that the sum of the first odd number is 1.
      Step 2: Inductive step: Assume that the sum of the first n odd numbers is n^2. Show that the sum of the first n + 1 odd numbers is (n + 1)^2.
      Step 3: Combine the two steps to complete the proof. *)
   using assms
proof (induct n)
   (* Step 1: Base case: Show that the sum of the first odd number is 1. *)
   case (Suc n)
   have ?case when "n=0" using that sledgehammer
   (* Step 2: Inductive step: Assume that the sum of the first n odd numbers is n^2. Show that the sum of the first n + 1 odd numbers is (n + 1)^2. *)
   moreover have ?case when "n>0"
   proof -
      have "(\<Sum>k = 0..(n-1). 2 * k + 1) = n^2" using Suc(1) that sledgehammer
      then have "(\<Sum>k = 0..(n-1). 2 * k + 1) + (2 * n + 1) = n^2 + 2 * n + 1" sledgehammer
      then show ?thesis sledgehammer
   qed
   (* Step 3: Combine the two steps to complete the proof. *)
   ultimately show ?case sledgehammer
qed simp
```

**Problem:**

For any natural number $n$ greater than or equal to 9, there exists a natural number $x$ such that $x^2$ is equal to the quotient of the difference between the factorial of $n + 2$ and the factorial of $n + 1$ divided by the factorial of $n$.

**Proof:**

```
theorem amc12b_2020_p6:
   fixes n :: nat
   assumes h0: "9 \<le> n"
   shows "\<exists>x::nat. (real_of_nat x)^2 = (fact (n + 2)
         - fact (n + 1)) / fact n"
```

*Prompt*

------------------------------------------------------------------------

*ChatGPT Output*

```
   (* Step 1: Use the definition of factorial to simplify the expression.
      Step 2: Rearrange the expression to obtain a perfect square.
      Step 3: Show that the perfect square is a natural number. *)
proof -
   (* Step 1: Use the definition of factorial to simplify the expression. *)
   have "(fact (n + 2) - fact (n + 1)) / fact n = ((n+2)*(n+1)*fact n - (n+1)*fact n) / fact n" using fact_reduce[of "n+1"] sledgehammer
   (* Step 2: Rearrange the expression to obtain a perfect square. *)
   also have "... = (n+1) * (n+2) - (n+1)" sledgehammer
   also have "... = (n+1)^2" sledgehammer
   (* Step 3: Show that the perfect square is a natural number. *)
   finally show ?thesis using assms sledgehammer
qed
```

Figure 15: A case from miniF2F-test. Our method recognizes viable subgoals and successfully finishes the proof.

As a mathematician familiar with Isabelle, your task is to provide a formal proof in response to a given problem statement.
Your proof should be structured and clearly written, meeting the following criteria:
- It can be verified by Isabelle.
- Each step of the proof should be explained in detail using comments enclosed in "(*" and "*)".
- The explanation for each step should be clear and concise, avoiding any unnecessary or apologetic language.
Please ensure that your proof is well-organized and easy to follow, with each step building upon the previous one.

**Problem:**
Prove that for any natural number $n$, 3 divides $n^3 + 2n$.

**Proof:**
```
theorem induction_divisibility_3divnto3m2n:
  fixes n::nat
  shows "3 dvd n^3 + 2 * n"
  (* Step 1: Prove the base case n = 0.
     Step 2: Assume the statement is true for some n and prove it for n + 1.
     Step 3: Expand (n + 1)^3 + 2(n + 1) and simplify.
     Step 4: Use the assumption that 3 divides n^3 + 2n to show that 3 divides 3n^2 + 3n + 3.
     Step 5: Combine steps 3 and 4 to show that 3 divides (n + 1)^3 + 2(n + 1).
     Step 6: Conclude that the statement is true for all natural numbers n. *)
proof (induct n)
  (* Step 1: Prove the base case n = 0.*)
  case 0
    then show ?case sledgehammer
  (* Step 2: Assume the statement is true for some n and prove it for n + 1.*)
  next case (Suc n)
  have ?case when "n=0"
    using that sledgehammer
  moreover have ?case when "n>0"
  proof -
  (* Step 3: Expand (n + 1)^3 + 2(n + 1) and simplify. *)
    have "(n + 1)^3 + 2 * (n + 1) = (n^3+3*n^2+3*n+1) + (2*n+2)"
      unfolding power3_eq_cube power2_eq_square sledgehammer
    also have "... = n^3+2*n + 3*n^2+3*n+3" sledgehammer
    finally have "(n+1)^3+2*(n+1) = n^3+2*n + 3*n^2+3*n+3".
  (* Step 4: Use the assumption that 3 divides n^3 + 2n to show that 3 divides 3n^2 + 3n + 3.*)
    moreover have "3 dvd n^3+2*n" using Suc(1) sledgehammer
    moreover have "3 dvd 3*n^2+3*n+3" sledgehammer
  (* Step 5: Combine steps 3 and 4 to show that 3 divides (n + 1)^3 + 2(n + 1). *)
    ultimately have "3 dvd (n+1)^3+2*(n+1)" sledgehammer
  (* Step 6: Conclude that the statement is true for all natural numbers n. *)
    then show ?thesis sledgehammer
  qed
  ultimately show ?case sledgehammer
qed
```

**Problem:**
Let $n$ be an integer and $p$ be a prime number. Show that $p$ divides $n$ if and only if $n^2$ is congruent to 0 modulo $p$.

**Proof:**
```
theorem numbertheory_prmdvsneqnsqmodpeq0:
  fixes n :: int
    and p :: nat
  assumes "prime p"
  shows "p dvd n \<longleftrightarrow> (n^2) mod p = 0"
  (* Step 1: Assume that p divides n. Show that n^2 is congruent to 0 modulo p.
     Step 2: Assume that n^2 is congruent to 0 modulo p. Show that p divides n. *)
proof -
  (* Step 1: Assume that p divides n. Show that n^2 is congruent to 0 modulo p.*)
  assume "p dvd n"
    then show "(n^2) mod p = 0" sledgehammer
  (* Step 2: Assume that n^2 is congruent to 0 modulo p. Show that p divides n. *)
  next assume "(n^2) mod p = 0"
    then show "p dvd n" sledgehammer
qed
```

**Problem:**
Suppose $n$ is a natural number such that $n$ leaves a remainder of 3 when divided by 7. Show that $(2n + 1)$ leaves a remainder of 0 when divided by 7.

**Proof:**
```
theorem mathd_numbertheory_370:
  fixes n :: nat
  assumes h0 : "n mod 7 = (3::nat)"
  shows "(2*n+1) mod 7 = (0::nat)"
```

*Prompt*

----------------------------------------------------------------------------------------

*ChatGPT Output*

```
  (* Step 1: Express n in terms of 7k + 3 for some k ∈ ℕ.
     Step 2: Substitute the expression for n into (2n + 1) and simplify.
     Step 3: Show that the result from Step 2 leaves a remainder of 0 when divided by 7. *)
proof -
  (* Step 1: Express n in terms of 7k + 3 for some k ∈ ℕ. *)
  have "n = 7 * (n div 7) + (n mod 7)" using div_mult_mod_eq[of "n" "7"] sledgehammer
  also have "... = 7 * (n div 7) + 3" using h0 sledgehammer
  finally have c0: "n = 7 * (n div 7) + 3".
  (* Step 2: Substitute the expression for n into (2n + 1) and simplify.*)
  have "(2*n+1) = 2*(7*(n div 7) + 3) + 1" using c0 sledgehammer
  also have "... = 14*(n div 7) + 7" sledgehammer
  also have "... = 7*(2*(n div 7) + 1)" sledgehammer
  finally have c1: "(2*n+1) = 7*(2*(n div 7) + 1)".
  (* Step 3: Show that the result from Step 2 leaves a remainder of 0 when divided by 7. *)
  have "(2*(n div 7) + 1) mod 7 = 0" using mod_mult_self2[of "2*(n div 7) + 1" "7"] sledgehammer
```
*Step error: Outer syntax error (line 1): command expected, but identifier error (line 1) was found At command \"<malformed>\" (line 1)*

Figure 16: A case from miniF2F-valid. The shuffled demonstration organization results in the incorrect application of the "$mod\_mult\_self2$" theorem.

As a mathematician familiar with Isabelle, your task is to provide a formal proof in response to a given problem statement.
Your proof should be structured and clearly written, meeting the following criteria:
- It can be verified by Isabelle.
- Each step of the proof should be explained in detail using comments enclosed in "(*" and "*)".
- The explanation for each step should be clear and concise, avoiding any unnecessary or apologetic language.
Please ensure that your proof is well-organized and easy to follow, with each step building upon the previous one.

**Problem:**

Suppose that $x$ and $y$ are two real numbers such that $x + y = 13$ and $xy = 24$. Find $\sqrt{x^2 + y^2}$ and show that it is equal to 11.

**Proof:**

```
theorem mathd_algebra_510:
  fixes x y :: real
  assumes h0 : "x+y=13"
    and h1 : "x*y=24"
  shows "sqrt (x^2 + y^2) = 11"
  (* Step 1: Use the identity x² + y² = (x + y)² − 2xy to find x² + y² in terms of x and y.
     Step 2: Substitute the given values of x and y to find x² + y² .
     Step 3: Show that x² + y² = 121.
     Step 4: Take the square root of both sides to find √(x² + y²).
     Step 5: Show that √(x² + y²) = 11. *)
proof -
  (* Step 1: Use the identity x² + y² = (x + y)² − 2xy to find x² + y² in terms of x and y. *)
  have c0: "x^2 + y^2 = (x+y)^2 - 2*x*y" sledgehammer
  (* Step 2: Substitute the given values of x and y to find x² + y² . *)
  also have "... = (13)^2 - 2*24" using h0 h1 sledgehammer
  (* Step 3: Show that x² + y² = 121. *)
  also have "... = 121" sledgehammer
  finally have c1: "x^2 + y^2 = 121" sledgehammer
  (* Step 4: Take the square root of both sides to find √(x² + y²). *)
  then show ?thesis sledgehammer
  (* Step 5: Show that √(x² + y²) = 11. *)
qed
```

**Problem:**

Suppose that $x$ is a positive real number. Show that $2 - \sqrt{2} \geq 2 - x - \frac{1}{2x}$.

**Proof:**

```
theorem algebra_amgm_faxinrrp2msqrt2geq2mxm1div2x:
  fixes x :: real
  assumes "x > 0"
  shows "2 - sqrt 2 \<ge> 2 - x - 1/ (2 * x)"
  (* Step 1: Show that √2 ≤ x + 1/2x.
     Step 2: Rearrange the inequality to get 2x² + 1 − 2x√2 ≥ 0.
     Step 3: Factor the expression to get (√2x − 1)² ≥ 0.
     Step 4: Conclude that the inequality is true. *)
proof -
  (* Step 1: Show that √2 ≤ x + 1/2x. *)
  have c0: "2 * x > 0" using assms sledgehammer
  have "sqrt 2 \<le> x + 1 / (2 * x)"
  proof -
    define y where "y = sqrt 2"
    have c1: "2 = y * y"
    proof -
      have "2 = (sqrt 2) * (sqrt 2)" sledgehammer
      then have "... = y * y" using \<open>y = sqrt 2\<close> sledgehammer
      then show ?thesis sledgehammer
    qed
    have "(2 * x) * x + 1 - (2 * x) * (sqrt 2) = (y * y * x * x) + 1 - (2 * x) * y" using c1 y_def sledgehammer
    also have "... = (y*x) * (y*x) - 2 * (y*x) + 1" sledgehammer
    also have "... = (y*x - 1) * (y*x - 1)" sledgehammer
    also have "... \<ge> 0" sledgehammer
    ultimately have c2: "(2 * x) * x + 1 - (2 * x) * (sqrt 2) \<ge> 0" sledgehammer
    have "(2*x) * (x + 1/(2*x) - sqrt 2) = (2 * x) * x + (2 * x) * (1/(2*x)) - (2*x) * sqrt 2" sledgehammer
    also have "... = (2 * x) * x + 1 - (2*x) * sqrt 2" using c0 sledgehammer
    (* Step 2: Rearrange the inequality to get 2x² + 1 − 2x√2 ≥ 0. *)
    also have "... \<ge> 0" using c2 sledgehammer
    (* Step 3: Factor the expression to get (√2x − 1)² ≥ 0. *)
    ultimately have "(2*x) * (x + 1/(2*x) - sqrt 2) \<ge> 0" sledgehammer
    hence "x + 1/(2*x) - sqrt 2 \<ge> 0" using mult_pos_pos sledgehammer
    (* Step 4: Conclude that the inequality is true. *)
    then show ?thesis sledgehammer
  qed
  then show ?thesis sledgehammer
qed
```

**Problem:**

Suppose that $x$ and $y$ are two real numbers such that $\frac{x+y}{2} = 7$ and $\sqrt{xy} = \sqrt{19}$. Find $x^2 + y^2$.

**Proof:**

```
theorem mathd_algebra_332:
  fixes x y ::real
  assumes "(x + y) / 2 = 7"
    and "sqrt (x * y) = sqrt 19"
  shows "x^2 + y^2 = 158"
```

*Prompt*

--------------------------------------------------------------------------------

*ChatGPT Output*

```
  (* Step 1: Use the identity x² + y² = (x + y)² − 2xy to find x² + y² in terms of x and y.
     Step 2: Substitute the given values of x and y to find x² + y².
     Step 3: Show that x² + y² = 158. *)
proof -
  (* Step 1: Use the identity x² + y² = (x + y)² − 2xy to find x² + y² in terms of x and y. *)
  have c0: "x^2 + y^2 = (x+y)^2 - 2*x*y" sledgehammer
  (* Step 2: Substitute the given values of x and y to find x² + y². *)
  also have "... = 2*7^2 - 2*19 " using assms sledgehammer
```

**Step error: Outer syntax error (line 1): command expected, but identifier error (line 1) was found At command \"<malformed>\" (line 1)**

Figure 17: A case from miniF2F-test. The shuffled demonstration organization leads to an unsuitable conjecture, "... = $27^2 − 219$," which cannot be proved using the Sledgehammer.

As a mathematician familiar with Isabelle, your task is to provide a formal proof in response to a given problem statement.
Your proof should be structured and clearly written, meeting the following criteria:
- It can be verified by Isabelle.
- Each step of the proof should be explained in detail using comments enclosed in "(*" and "*)".
- The explanation for each step should be clear and concise, avoiding any unnecessary or apologetic language.
Please ensure that your proof is well-organized and easy to follow, with each step building upon the previous one.

**Problem:**

Given a real number $n$ such that $n \neq 3$ and $\frac{(n+5)}{(n-3)} = 2$, show that $n = 11$.

**Problem:**

Prove by induction that the sum of the first $n$ odd numbers is $n^2$.

**Proof:**

```
theorem induction_sum_odd:
    fixes n :: nat
    assumes "n > 0"
    shows "(\<Sum> (k::nat) = 0..(n-1). 2 * k + 1) = n^2"
    (* Step 1: Base case: Show that the sum of the first odd number is 1.
       Step 2: Inductive step: Assume that the sum of the first n odd numbers is n². Show that the sum of the first n + 1 odd numbers is (n + 1)².
       Step 3: Combine the two steps to complete the proof. *)
    using assms
proof (induct n)
    (* Step 1: Base case: Show that the sum of the first odd number is 1. *)
    case (Suc n)
    have ?case when "n=0" using that sledgehammer
    (* Step 2: Inductive step: Assume that the sum of the first n odd numbers is n². Show that the sum of the first n + 1 odd numbers is (n + 1)². *)
    moreover have ?case when "n>0"
    proof -
        have "(\<Sum>k = 0..(n-1). 2 * k + 1) = n^2" using Suc(1) that sledgehammer
        then have "(\<Sum>k = 0..(n-1). 2 * k + 1) + (2 * n + 1) = n^2 + 2 * n + 1" sledgehammer
        then show ?thesis sledgehammer
    qed
    (* Step 3: Combine the two steps to complete the proof. *)
    ultimately show ?case sledgehammer
qed simp
```

**Proof:**

```
theorem mathd_algebra_181:
    fixes n :: real
    assumes h0 : "n \<noteq> 3" and h1 : "(n+5) / (n-3) = 2"
    shows "n=11"
    (* Step 1: Use the given equation (n+5)/(n-3) = 2 to obtain an equation involving n.
       Step 2: Simplify the equation to obtain an expression for n.
       Step 3: Show that the expression for n is equal to 11. *)
proof -
    (* Step 1: Use the given equation (n+5)/(n-3) = 2 to obtain an equation involving n. *)
    have "n+5 = 2 * (n-3)" using h0 h1 sledgehammer
    (* Step 2: Simplify the equation to obtain an expression for n. *)
    thus ?thesis sledgehammer
    (* Step 3: Show that the expression for n is equal to 11. *)
qed
```

**Problem:**

For any natural number $n$ greater than or equal to 9, there exists a natural number $x$ such that $x^2$ is equal to the quotient of the difference between the factorial of $n + 2$ and the factorial of $n + 1$ divided by the factorial of $n$.

**Proof:**

```
theorem amc12b_2020_p6:
    fixes n :: nat
    assumes h0: "9 \<le> n"
    shows "\<exists>x::nat. (real_of_nat x)^2 = (fact (n + 2)
        - fact (n + 1)) / fact n"
```

*Prompt*

---

*ChatGPT Output*

```
    (* Step 1: Simplify the expression on the right-hand side of the equation using the definition of factorial.
       Step 2: Rearrange the equation to obtain an expression for x².
       Step 3: Show that the expression for x² is a perfect square.
       Step 4: Conclude that there exists a natural number x such that x² is equal to the given expression. *)
proof -
    (* Step 1: Simplify the expression on the right-hand side of the equation using the definition of factorial. *)
    have "(fact (n + 2) - fact (n + 1)) / fact n = ((n + 2) * fact (n + 1) - fact (n + 1)) / fact n" by (simp add: fact_reduce)
```

*Step error: Failed to finish proof (line 1): goal (1 subgoal):*
*1. (2 + real n) * ((1 + real n) * fact n) - (1 + real n) * fact n = fact n + real n * fact n + real n * (fact n + real n * fact n)*
*At command \"by\" (line 1)*

Figure 18: A case from miniF2F-test. The shuffling of the optimal demonstration organization leads to a complex and unresolved equation, which further results in a failure to properly simplify the factorial expression.

