# OpenReview forum: "Decomposing the Enigma: Subgoal-based Demonstration Learning for Formal Theorem Proving"
_ICLR.cc/2024/Conference — Submitted to ICLR 2024_

### Official Review · Reviewer_kVAV · 2023-11-01

**Soundness:** 3 good
**Presentation:** 3 good
**Contribution:** 3 good
**Rating:** 8
**Confidence:** 2

**Summary:**

This work aims at using LLMs to assist formal proof generation. Previous attempts decompose the generation process into two phases, where the LLMs first generate an informal proof given the statement and then build a formal proof given the informal proof. The authors argue that LLM-generated informal proofs are prone to be invalid while human-written ones could be incompatible with the formal proof. To bridge the gap, the authors propose to replace the informal proof with a sequence of subgoal proofs, where each step of the subgoals can be verified. The subgoal proofs are iteratively generated with LLMs and existing automated theorem prover as the verifier.

The authors also claim that the order and selection of demonstrations are vital to the performance of LLM-based generations. To automate this selection process, the authors utilize a diffusion model to generate conditional demonstration organizations.

Empirically, the proposed method can outperform previous baselines in the MiniF2F dataset or match the performance while significantly reducing the number of LLM calls.

**Strengths:**

- The paper is well-written and easy to follow.
- The idea of using subgoal-based proofs to bridge the gap between formal proofs and LLMs is well-motivated.
- There is a clear empirical improvement over previous baselines.

**Weaknesses:**

- For the diffusion model-based reorganization, it would be nice to show the correlation between the generated optimal demonstration organization and the input statement. I am curious if the diffusion model would collapse into generating a generic organization that works decently well with most statements. On roughly the same note, it would also be nice to show some examples where the order of the demonstrations greatly affects the performances. Another interesting baseline to compare would be to search the optimal organization for each tested statement.
- It seems that previous LLM-based algorithms do not utilize formal proof generation tools such as Sledgehammer. I am wondering if it's possible to add a comparison of the number of usages of such tools (or their resource consumptions), aligning with the LLM calls comparison.

**Questions:**

I don't have any questions for the authors.

---

> ### Author Response · Authors · 2023-11-19
> **Response to Reviewer kVAV**
>
> We sincerely appreciate your time and constructive suggestions, particularly your recognition of our work, which has been greatly encouraging. We hope our responses adequately address all of your concerns.
>
> **Question 1: Whether the diffusion model generates a generic organization effective for most statements.**
>
> We deeply appreciate your insightful advice regarding the diffusion model's performance. Our empirical finding strongly suggests that there isn't a “one-size-fits-all” optimal demonstration organization. Specifically, even the most adaptable organization we identified could successfully prove only *3* distinct statements. Furthermore, we only observed *4* instances of such adaptable organizations on the miniF2F-test. This highlights the tailored nature of each organization to its corresponding statement. The limited scope of even the most adaptable organization indicates the challenge in identifying the optimal demonstration organization for each unique statement. This complexity further emphasizes the effectiveness of our model in efficiently searching for and determining the most suitable demonstration organization for individual statements. These discussions are now included in Appendix D of the updated manuscript.
>
> **Question 2: Examples about how demonstration order affects performance.**
>
> We are sincerely grateful for your valuable suggestion and have taken steps to address it. We have added several illustrative examples in Figures 16-18 in Appendix E. These cases specifically highlight how the order of demonstrations can significantly impact the final performance.
>
> **Question 3: Comparing with a baseline that searches for optimal organization for each tested statement.**
>
> Thank you for your constructive suggestion. We conducted additional experiments using 200 randomized organization sampling attempts for each statement in the miniF2F-valid and miniF2F-test to search for the optimal organization. The results are as follows:
>
> | Method                       | Valid  | Test  |
> |------------------------------|--------|-------|
> | Ours                         | 48.0%  | 45.5% |
> | Optimal Organization Search  | 48.4%  | 44.3% |
>
> As noted in [1], the benefit of conducting extensive organization searches tends to diminish beyond 100 attempts. This finding highlights the inherent challenges presented by this dataset and indicates the effectiveness of our diffusion model in efficiently optimizing the search process, thereby conserving costs in using LLMs. These results have been incorporated into Appendix C.7 in the updated version of our manuscript.
>
> **Question 4: Comparing resource consumption of Sledgehammer in LLM-based algorithms.**
>
> Thank you for your valuable feedback! We have further analyzed the average number of Sledgehammer calls and their execution times, comparing our method with the DSP algorithm[1]. This analysis was conducted on a machine equipped with 64 CPU cores.
> It is important to note that DSP also uses Sledgehammer. We have clarified this point more explicitly in the updated version. The results are as follows, displaying the average number of Sledgehammer calls and their corresponding durations (in seconds) for each solved statement:
>
> | **Method** | **Calls (valid)** | **Calls (test)** | **Duration (valid)** | **Duration (test)** |
> |------------|-------------------|------------------|----------------------|---------------------|
> | DSP        | 2.33              | 2.39             | 3.29                 | 2.98                |
> | Ours       | 2.88              | 3.22             | 4.16                 | 4.94                |
>
>
> These results show that our method has a slight increase in both the frequency of Sledgehammer calls and their execution times compared to DSP. Specifically, this increase is primarily observed in statements that our method can solve but DSP cannot. For these statements, the number of Sledgehammer calls on the miniF2F-valid and miniF2F-test are 3.21 and 5.38, respectively. This suggests that the need for Sledgehammer becomes more important as the problem complexity increases. These results have been incorporated into Appendix C.8 in the updated version of our manuscript.
>
> These results indicate that optimizing the use of Sledgehammer could potentially enhance the effectiveness of existing LLM-based methods. However, it is important to note that such an optimization strategy falls beyond the scope of our current paper. Nonetheless, your suggestion has sparked interest in this as a valuable direction for future research.
>
> **Reference**:
>
> [1] Jiang A Q, Welleck S, Zhou J P, et al. Draft, sketch, and prove: Guiding formal theorem provers with informal proofs. ICLR 2023.

---

> > ### Comment · Reviewer_kVAV · 2023-11-20
> >
> > Thank you for the response. I believe my concerns have been addressed.

---

### Official Review · Reviewer_Wmqb · 2023-11-01

**Soundness:** 3 good
**Presentation:** 3 good
**Contribution:** 3 good
**Rating:** 6
**Confidence:** 3

**Summary:**

This paper looks at using LLMs to provide formal proofs from an initial human-provided proof sketch, where the formal proofs are verifiable by an external automated theorem prover. The proposed method consists of two parts. First, the authors propose generating subgoal based proofs from human-provided proof sketches. This is inspired by the subgoal generation literature in reinforcement learning--specifically, that a good sequence of subgoals should preserve the property that each subgoal is reachable from the start state, and that the goal state is reachable from the subgoal. Second, the authors propose using diffusion models to select and order the demonstrations provided to the LLMs, for generating the formal theorem sketches.

**Strengths:**

- Using LLMs for theorem proving is an interesting problem. The proposed method follows prior work in breaking down the problem into two in-context  learning problems: first, providing a subgoal proof, and second, providing a formal proof based on the subgoal proof. The paper proposes novel methods to order in-context learning examples for the in-context learning problems.
- Well written and clear.
- Experiments are described clearly and sufficient baselines are used. Experiments show ~7% improvement of proposed method over baselines.

**Weaknesses:**

Experiments:
- No std deviations or measurement of uncertainty in Tbl 1 &2.
- The improvement brought by using the diffusion models is very minor (about 1% in the Table 2). This is potentially not statistically significant. However, in Figure 2a, there seems to be a clear  advantage of using diffusion models + subgoals over using subgoals alone (about 2-5 additional problems solved for each # of LLM calls). These two results seem slightly contradictory. In any case, the result of Table 2 shows that the main innovation of the method seems to be the subgoal generation itself, which I feel the authors should state clearly.

Experimental setup is not fully justified.
- Why are words like "sorry" and "oops" used to determine that the proof is not valid? The provided justification is that these words indicate that the proof has been prematurely terminated, but I'm not quite sure I understand this justification.

**Questions:**

How does the quality of the human provided proof sketch influence the performance of the proposed method?

---

> ### Author Response · Authors · 2023-11-19
> **Response to Reviewer Wmqb**
>
> Thank you for dedicating your time and providing constructive suggestions. We hope our responses can thoroughly address your concerns.
>
> **Question 1: Lack of standard deviation and uncertainty measures.**
>
> Thank you for your valuable suggestions! We have included additional results with mean and standard deviation values to quantify the uncertainty associated with our method. Specifically, for the variants employing diffusion models (”Ours” and “-subgoal”), we trained three separate diffusion models as detailed in Section 3.4. For the variants without diffusion models (”-subgoal & diffusion” and “-diffusion”), we repeated the experiment three times. The results are as follows and we have also updated these values in Table 2 of our manuscript:
>
> | Method               | Valid             | Test              |
> |----------------------|-------------------|-------------------|
> | Ours                 | 48.0% (*±0.4*)    | 45.5% (*±0.6*)    |
> | -subgoal & diffusion | 41.4% (*±0.9*)    | 38.7% (*±1.2*)    |
> | -subgoal             | 44.3% (*±0.7*)    | 40.6% (*±0.6*)    |
> | -diffusion           | 46.9% (*±1.3*)    | 44.1% (*±0.9*)    |
>
> These results demonstrate that the methods incorporating diffusion models exhibit relatively more stable performance.
>
> For baseline methods in Table 1, we used reported results from their original papers to ensure a fair comparison, hence standard deviations for these are not included in our analysis.
>
> **Question 2: Inconsistency in diffusion model's efficacy.**
>
> We greatly appreciate your perceptive remarks regarding the diffusion model's performance in our study. As detailed in Section 4.3, we found that the benefit of the diffusion model tends to decrease when the number of Large Language Model (LLM) calls reaches 100. This pattern is attributed to the model being trained on data points where optimal demonstration organization is achievable through a reasonable number of randomized organization sampling attempts. Consequently, it is adept at generalizing to *unseen* data points of similar complexity, thereby effectively reducing the number of LLM calls required for these *unseen* scenarios. This capability notably results in the more pronounced gap between “subgoal+diff” and “subgoal” observed in Figure 2a, particularly evident when the LLM calls are reduced to 20.
>
> On the other hand, the model's efficiency in organizing demonstrations may not be as marked for data points demanding a significantly higher count of randomized sampling attempts for optimal organization. This factor contributes to the relatively limited effect of the diffusion model shown in Table 2 (It is important to highlight that the results presented in Table 1 & 2 were obtained with 100 LLM calls, aligning with the standard experimental setups prevalent in the field [1]).
>
> **Question 3: Clarification on the use of "sorry" and "oops" to determine invalid proofs.**
>
> In the Isabelle proof assistant, "sorry" can be interpreted as "we presume the current proposition is true (without proving it)", and "oops" means that "we give up showing the current proposition". For example, both `lemma "1=(2::int)" sorry` and `lemma "1=(2::int)" oops` are valid Isabelle code: the first assumes the integers 1 and 2 are equal (and this is treated as a lemma that can then be used in other proofs); in the second case, we abandon the attempt to show 1=2. We, of course, will not accept either as a valid proof of the equation 1 = 2. Therefore, any proof that includes the terms "oops" or "sorry" will be deemed invalid.
>
> **Question 4: Impact of human-provided proof sketch quality on method performance.**
>
> Thank you for this valuable advice in making the evaluation of our method more solid. We additionally used gpt-3.5-turbo-0613 to generate informal proofs for problems in the miniF2F-valid. We applied the algorithms detailed in Section 2.1 to construct subgoal-based proofs. A total of 61 problems were then selected as demonstration samples for our experiments, in accordance with the setup described in Section 3.4. The results obtained are as follows:
>
> | Method                        | Valid            | Test             |
> |-------------------------------|------------------|------------------|
> | Ours                          | 48.0% (±0.4)     | 45.5% (±0.6)     |
> | Ours (GPT-3.5 Informal Proofs)| 47.7% (±0.5)     | 45.0% (±0.7)     |
>
> These results suggest that our method demonstrates robustness against variations in the quality of informal proof sketches. These additional results have been detailed in Appendix C.6 of the updated version of our manuscript.
>
> **Reference**:
>
> [1] Jiang A Q, Welleck S, Zhou J P, et al. Draft, sketch, and prove: Guiding formal theorem provers with informal proofs. ICLR 2023.

---

> ### Author Response · Authors · 2023-11-22
> **Further comments and discussions will be appreciated!**
>
> Dear Reviewer Wmqb,
>
> Thank you for your time and effort in reviewing our manuscript. We greatly appreciate your insightful comments, which have been invaluable in enhancing the quality of our work.
>
> In response:
>
> 1. We have enriched our manuscript with additional experimental results to better quantify uncertainty and evaluate the impact of human-provided proof sketch quality.
>
> 1. Additionally, we have provided detailed explanations regarding the inconsistency in the diffusion model's efficacy and the use of terms such as "sorry" and "oops" in determining invalid proofs.
>
> We hope our answers have addressed your concerns. If you have any further questions, we are happy to address them. We would really appreciate it if you are willing to reconsider your score.
>
> Thanks very much!
>
> Best regards,
>
> Authors

---

### Official Review · Reviewer_xitB · 2023-11-11

**Soundness:** 2 fair
**Presentation:** 2 fair
**Contribution:** 2 fair
**Rating:** 5
**Confidence:** 3

**Summary:**

This paper presents a method of improving the performance of LLM-based guidance to the formal prover Isabell for the Isabell problems in MiniF2F.  The reported results are an improvement on previous results for this dataset.  The primary method involves the constructions of in-context examples to serve as a prompt to an LLM in finding a proof of a desired theorem x.  The prompt consists of a set of "demonstrations" each of which is a kind of "lemma" consisting of the statement proved by the lemma and the sequence of steps (subgoals) in the proof of the lemma. The method of constructing the demonstrations in the prompt appropriate for a specific goal x is quite complex involving a heuristic solution to a Hamiltonian graph problem.

**Strengths:**

This paper described an considerable effort in improving the performance of LLM-based support for formal verification.  The ideas are nontrivial and provide experimental results that may be of value to future efforts in this area.

**Weaknesses:**

The writing is unclear.  For example does the term "demonstration-based proof" in the subgoal refinement section correspond to graph nodes (demonstrations) in section 2.2?  Reviewers cannot be expected to read appendices and the technical meaning of terms needs to be clear from the body of the paper.

A serious ambiguity is whether the "manually written" seed demonstration-based proofs are biased to the test set.  If so, this would constitute training on the test set.  The process of manual annotation needs to be discussed.  A related issue is source of the training data for the "diffusion model" (it is not really a diffusion model as the latent variables are not in R^d and no Gaussian distributions are involved).
In the section on evaluation they say

   Given the absence of a training split in the miniF2F
   dataset, we leverage optimal organizations that yield successful proofs from the miniF2F-valid set
   to train the diffusion model.

I find this very unclear.  Why is training on the validation set ok?

It seems to me that the complexity of the system allows for an over-fitting of the system design to the test data. I worry that the results would not generalize to a test set that was unavailable to the authors before testing.

**Questions:**

Questions are asked in the weakness section.

---

> ### Author Response · Authors · 2023-11-19
> **Response to Reviewer xitB (Part 1/2 )**
>
> Thank you for your insightful feedback and for recognizing the effort and novelty of our work. We appreciate the opportunity to clarify the points you have raised and address your concerns.
>
> **Question 1: Unclear definition of “demonstration-based proof” in the subgoal refinement section and its relation to graph nodes in section 2.2.**
>
> We appreciate your query and apologize for any confusion. We would like to gently clarify that the correct term we intended to use is “subgoal-based proof”, as opposed to “demonstration-based proof”.
>
> The term “subgoal-based proof” is used to describe the in-context examples that are generated during the subgoal refinement process, as detailed in Section 2.1. Each of these examples includes a mathematical statement, a proof based on subgoals, and a corresponding formal sketch, as introduced in the opening paragraph of Section 2. Additionally, we would like to clarify that these subgoal-based proofs indeed serve as the graph nodes, as highlighted in the first paragraph of Section 2.2. We are grateful for the opportunity to clarify and hope this clarification better illustrates the connection between the concepts.
>
> **Question 2: Potential bias in the manually written seed demonstration-based proofs.**
>
> We understand the criticality of this concern. To mitigate such biases, we sourced the manually written seeds from DSP’s informal proofs [1], which are entirely independent of the test set. The selection was based solely on the success of constructing a subgoal proof, not on any similarity or relevance to the test set. It also makes our results comparable with the DSP paper.
>
> **Question 3: Training data for the diffusion model.**
>
> In our study, we focus on enhancing the efficacy of Large Language Models (LLMs) in formal theorem proving. A key aspect of our experiment is the use of the miniF2F dataset, a benchmark where almost no formal proofs are provided (in neither miniF2F-valid nor miniF2F-test) and the goal is to supplement those statements with formal proofs. This makes it suitable for evaluating LLMs as no formal proof of the valid/test questions can be found online.
> Neither the previous work [1][2] nor this work has formal proofs to train on.
> To make our approach performance-wise comparable to prior work like DSP [1], which effectively used the miniF2F-valid to guide LLMs in proof generation for miniF2F-test problems, we retained the same set-up. Concretely, we use the statements from the validation set to obtain better prompts (i.e., demonstrations) that generate better informal proofs (i.e., subgoal-based proofs) to guide formal theorem proving. As detailed in Section 3.4, we further divided the miniF2F-valid into non-overlapping training and validation subsets. This division allowed us to conduct hyperparameter tuning and implement early stopping mechanisms in the validation subset. This should ensure a fair and reliable evaluation of our model performance in the miniF2F test set.
>
> **Question 4: Clarification of the definition of “diffusion Model” used in the study.**
>
> We greatly appreciate your observation regarding the nature of the diffusion model employed in our framework. It is important to clarify that our model is indeed a discrete diffusion model incorporating Bernoulli noise. This approach is in accordance with the methodologies described in [4].
>
> **Question 5: The potential lack of generalizability to an unseen test set.**
>
> We appreciate the chance to highlight that our approach has shown promising results with in-domain *unseen* test data (”in-domain” means that all problems come from high-school level practices or competitions). As we elaborated in our response to **Question 3**, the miniF2F-test was kept completely *unseen* during the hyperparameter tuning process, ensuring an unbiased evaluation of our model.
>
> We acknowledge that our current diffusion model may face challenges in transferring to out-of-domain data, such as undergraduate-level problems. Our methodology has the potential for wider application as more diverse datasets, such as ProofNet [3] (currently available in Lean with an Isabelle version in progress), become available. The current scope of our research has successfully demonstrated that with a judicious use of annotated data, our learned diffusion model can achieve competitive results with only 1/5 LLMs calls on the *unseen* miniF2F-test, indicating the potential of our approach to adapt to more diverse training set and generalize to out-of-domain test sets in future work. We are grateful for the opportunity to clarify this aspect of our study and thank you for bringing this important topic to our attention. We look forward to contributing further to this exciting field of research.
>
> We sincerely appreciate the opportunity to address the concerns raised and to further clarify the contributions of our work. Should there be any need for additional clarifications, we are more than willing to provide them.

---

> > ### Author Response · Authors · 2023-11-19
> > **Response to Reviewer xitB (Part 2/2 )**
> >
> > **References**:
> >
> > [1] Jiang A Q, Welleck S, Zhou J P, et al. Draft, sketch, and prove: Guiding formal theorem provers with informal proofs. ICLR 2023.
> >
> > [2] Lample G, Lacroix T, Lachaux M A, et al. Hypertree proof search for neural theorem proving. NeurIPS 2022.
> >
> > [3] Azerbayev Z, Piotrowski B, Schoelkopf H, et al. Proofnet: Autoformalizing and formally proving undergraduate-level mathematics. arXiv preprint.
> >
> > [4] Austin J, Johnson D D, Ho J, et al. Structured denoising diffusion models in discrete state-spaces. NeurIPS 2021.

---

> ### Comment · Reviewer_xitB · 2023-11-19
> **Clarifications**
>
> Rather than try to extract clarity from the paper, which I am fining very difficult, I'll just ask.
>
> I'm still having trouble understanding your basic data structures. Your (informal) presentation seems ambiguous (as informal statements tend to be, which is the subject of your paper!).  I'm trying to "auto-formalize" your paper.
>
> You seem to be saying that a subgoal-based proof is a pair <conclusion, {premise1,...,premiseK}> where the conclusion follows logically from the premises.  Is that correct?  Has each proof been formally verified? If so wouldn't it be much clearer to say this explicitly? What language are these statements written in.  What syntactic category of Isabell expressions are we talking about?

---

> > ### Author Response · Authors · 2023-11-20
> > **Further Clarifications**
> >
> > Thank you for your continued engagement and insightful questions!
> >
> > **Question 1: Further clarifications about the data structures.**
> >
> > To better illustrate the data structures discussed in our paper, specifically the terms "statement", "subgoal-based proof" and "formal sketch", we present the following example (also shown in Figure 3 of our paper):
> >
> > ```
> > - Statement:
> >
> >   Suppose n is a natural number such that n leaves a remainder of 3 when divided by 7. Show that (2n+1) leaves a remainder of 0 when divided by 7.
> >
> > - Subgoal-Based Proof:
> >
> >   Step 1: Express n as 7k+3 for some k \in N.
> >   Step 2: Substitute the expression for n into the expression for 2n+1 and simplify.
> >   Step 3: Show that the resulting expression is divisible by 7.
> >
> > - Formal Sketch:
> >   theorem
> >       fixes n :: nat
> >       assumes h0 : "n mod 7 = (3::nat)"
> >       shows "(2*n+1) mod 7 = (0::nat)"
> >   proof -
> >       (* Step 1: Express n as 7k+3 for some k \in N. *)
> >       have "n = 7 * (n div 7) + (n mod 7)"
> >           using div_mult_mod_eq[of "n" "7"] sledgehammer
> >       also have "... = 7 * (n div 7) + 3”
> >           using h0 sledgehammer
> >       finally have c0: "n = 7 * (n div 7) + 3".
> >       (* Step 2: Substitute the expression for n into the expression for 2n+1 and simplify. *)
> >       have "2*n+1 = 2*(7*(n div 7) + 3) + 1"
> >           using c0 sledgehammer
> >       also have "... = 14*(n div 7) + 7”
> >           sledgehammer
> >       also have "... = 7*(2*(n div 7) + 1)"
> >           sledgehammer
> >       finally have c1: "2*n+1 = 7*(2*(n div 7) + 1)” .
> >       (* Step 3: Show that the resulting expression is divisible by 7. *)
> >       show "(2*n+1) mod 7 = 0"
> >           using c1 sledgehammer
> >   qed
> >
> > ```
> >
> > An in-context example (or graph node) follows the structure of including the statement, the subgoal-based proof, and the formal sketch. A subgoal-based proof consists of a series of subgoals written in text. In our previous response, we described subgoal-based proofs as integral components of the graph nodes. However, to clarify, it’s more accurate to state that it is the in-context examples that serve as graph nodes. Each subgoal-based proof is verified to successfully guide formal theorem proving. In this work, the subgoals are presented in text, while the formal sketch is expressed in the Isabelle language.
> >
> > Thank you once again for your valuable guidance in this important clarification, and we have made it clear in our updated manuscript.
> >
> > **Question 2: What syntactic category of Isabell expressions are we talking about?**
> >
> > Thank you for your question regarding the syntactic categories of Isabelle expressions used in our paper.
> >
> > We have interpreted your query to pertain to the distinction between "inner syntax" and "outer syntax" in Isabelle, which are the primary categories we've employed:
> >
> > - **Inner Syntax**: Involves logical statements and propositions such as theorems, lemmas, and assumptions, typically using standard logical connectives and quantifiers.
> >
> > - **Outer Syntax**: Comprises commands that structure and advance the proof, including directives like `have`, `show`, and `thus`.
> >
> > However, we wish to ensure that we are addressing your query accurately. If your question refers to a different aspect of syntactic categorization in Isabelle or a more specific context within our work, could you kindly provide additional details or clarification? We aim to respond as precisely and helpfully as possible.

---

> ### Author Response · Authors · 2023-11-22
> **Further comments and discussions will be appreciated!**
>
> Dear Reviewer xitB,
>
> Thank you for your constructive feedback on our manuscript. Your insights have been invaluable in guiding our revisions. In response, we have included detailed clarifications regarding the data structure and the syntactic category of Isabelle expressions. We hope our answers have addressed your concerns. If you have any further questions, we are happy to address them. We would really appreciate it if you are willing to reconsider your score.
>
> Thanks very much!
>
> Best regards,
>
> Authors

---

### Author Response · Authors · 2023-11-19
**General response to all reviewers**

We would like to express our gratitude to the reviewers for their detailed and encouraging feedback. We appreciate their recognition of the importance of our research topic, the novelty of our method, the clarity of our writing, the substantial improvements we have made, and the potential impact on future research.

We carefully addressed the specific comments and suggestions provided by each reviewer, as outlined in our individual responses. We believe that we have addressed all concerns and are here ready to provide further information.

---

### Meta-Review · Area_Chair_Pb4A · 2023-12-14

**Metareview:**

This paper addresses the use of LLMs in finding formal proofs. Results are reported for the MiniF2F benchmark. The method inserts "subgoal based proofs" into the context. A subgoal based proof consists of a set of "demonstrations" each of which is a kind of "lemma" consisting of the statement proved and the sequence of steps (subgoals) in the proof of the lemma. The method of constructing the subgoal based proof involves a heuristic solution to a Hamiltonian graph problem.

Several reviewers initially assigned to this paper declined saying they could not evaluate it.  They could not read the paper as the paper is very opaque and difficult to understand.  One of the reviewers tried to get clarity in the terminology used and the response was to send code rather than give an appropriate definition of terms.

**Justification For Why Not Higher Score:**

I feel that the paper is too poorly written, and the ideas too poorly defined, to appear at ICLR.

**Justification For Why Not Lower Score:**

That is the lowest score.

---

### Decision · Program_Chairs · 2024-01-16

Reject